# Biallelic variants in *RNU2-2* cause a remarkably frequent developmental and epileptic encephalopathy

Neurodevelopmental disorders (NDDs) affect 2–4% of the population, are predominantly genetic and remain unsolved in ~50% of individuals. We show that rare biallelic variants in *RNU2-2* are enriched and over-transmitted in individuals with unresolved NDDs. We define a recessive *RNU2-2* syndrome, delineate its unique genetic architecture and show that it manifests clinically as a severe developmental and epileptic encephalopathy. We find that candidate biallelic variants are significantly correlated with reduced U2-2 abundance, implicating compromised transcript stability as a probable pathomechanism. We identify a decreased ratio of U2-2 to its paralog U2-1 as a potential diagnostic biomarker for this condition. We show that the recessive *RNU2-2* syndrome is genetically, clinically and mechanistically distinct from the dominant *RNU2-2* disorder. Within our cohort, the recessive *RNU2-2* syndrome emerges as by far the most frequent recessive NDD, greatly disproportionate to the small genomic footprint of this non-protein-coding gene.

NDDs are heterogeneous conditions that affect 2–4% of the population[1,2]. They are frequently genetic, and accurate diagnosis underpins their clinical management, genetic counseling and reproductive decision-making[3]. Even with genome sequencing, almost half of NDD cases remain unsolved[4,5]. Variants in several small nuclear RNA (snRNA) genes cause human genetic conditions, including NDDs (Supplementary Table 1).

Spliceosomes are snRNAs containing complexes that are essential for splicing[6]. U1, U2, U4 and U6 snRNAs are incorporated into the major spliceosome, while equivalent U11, U12, U4atac and U6atac participate in the minor spliceosome. U5 functions in both spliceosomes[7]. The U2 snRNA recognizes the splicing branch site through RNA–RNA base pairing during spliceosomal assembly and facilitates nucleophilic attack of the branchpoint adenosine at the 5′ splice site in intron lariat formation[8]. U2 comprises 191 nucleotides and has two known, nearly identical functional paralogs: U2-1 and U2-2. U2-1 is encoded from a 6.1 kb tandem repeat array of five to 82 identical copies of *RNU2-1* on human chromosome 17 (ref. 9). U2-2 is encoded by single-exon gene *RNU2-2*, located on chromosome 11. Recently, we and others discovered a frequent dominant NDD caused by heterozygous *RNU2-2* variants[10,11].

Here, we describe a highly prevalent NDD and developmental and epileptic encephalopathy (DEE) caused by biallelic *RNU2-2* variants. We show that rare biallelic *RNU2-2* variants are enriched and over-transmitted in a cohort of individuals with unsolved NDDs. We provide compelling evidence that these variants cause a recessive DEE with a distinctive genetic architecture. We show that candidate variants are associated with U2-2 transcript depletion and a decreased U2-2:U2-1 transcript ratio. We also show that the dominant and recessive *RNU2-2* disorders are genetically, clinically and mechanistically distinct. Finally, we demonstrate that the recessive *RNU2-2*-related disorder is the most common recessive NDD in our cohort.

## Results

### Biallelic *RNU2-2* variants are enriched and over-transmitted in individuals with unsolved NDD

We reasoned that snRNAs are promising candidates for novel recessive NDDs. We conducted an enrichment analysis for biallelic variants in snRNA genes using aggregated short-read genome sequencing data with statistical phasing from 78,051 individuals in the 100,000 Genomes Project (100kGP)[12,13]. We quantified the frequency of rare (internal

✉e-mail: adam.jackson@manchester.ac.uk; siddharth.banka@manchester.ac.uk

minor allele frequency of <0.001) homozygous and/or compound heterozygous variants for all 1,901 snRNAs annotated in GENCODE v.32. In total, we identified 1,897 homozygous and 1,692 compound heterozygous variants in 2,486 individuals. We compared the frequency of rare biallelic variants in individuals with unsolved NDD ($n$ = 6,762) versus all other individuals in this cohort (henceforth referred to as '100kGP controls'; $n$ = 71,289). The 100kGP controls included unaffected relatives, individuals with non-NDD phenotypes and individuals with previously solved NDD. We identified nominal enrichment for biallelic variants in individuals with unsolved NDD in two out of three snRNAs genes known to be associated with recessive disorders, including *RNU4-2* (refs. 14,15) (odds ratio (OR) = 5.75, 95% CI = 1.75–17.0, two-sided Fisher's exact test $P$ = 0.00225) and *RNU12* (refs. 16,17) (OR = 3.84, 95% CI = 1.48–8.95, $P$ = 0.00325) but not *RNU4ATAC*[18] (OR = 1.56, 95% CI = 0.642–3.31, $P$ = 0.252) (Fig. 1a and Supplementary Table 2). We also identified significant enrichment for rare biallelic genotypes in *RNU2-2* (NR_199791.1) (all variants: OR = 4.40, 95% CI = 2.81–56.72, two-sided Fisher's exact test $P$ = 4.4 × 10⁻¹⁰; homozygous variants: OR = 9.94, 95% CI = 4.70–20.93, $P$ = 2.74 × 10⁻⁹; compound heterozygous variants only: OR = 2.82, 95% CI = 1.51–4.95, $P$ = 7 × 10⁻⁴).

In a rare conditions cohort, for recessive conditions, affected individuals are expected to inherit two pathogenic alleles from carrier parents more frequently than the expected one in four chance[19]. Using genome sequencing data from 12,015 trios in the 100kGP, we identified 49 trios in which both father and mother were heterozygous for rare *RNU2-2* variants. In total, 33 out of 49 offspring from these trios were probands with unsolved NDD (18 homozygotes and 15 compound heterozygotes). We detected a significant over-transmission of *RNU2-2* variants from parents to probands with unsolved NDD versus the expected Mendelian ratios (chi-squared goodness-of-fit, $P$ = 1.97 × 10⁻⁶) (Fig. 1b). For the remaining 16 trios in which the proband did not have an unsolved NDD, there was no significant deviation from the expected Mendelian ratio for any combination of variants (Extended Data Fig. 1).

Collectively, the results suggest *RNU2-2* is a likely recessive NDD gene.

### Read-based phasing identifies biallelic *RNU2-2* variants

From the statistically phased sequencing data described above, we identified 223 individuals with two or more rare *RNU2-2* variants (33 homozygous and 190 individuals with two different variants: 76 phased in *trans* and 114 phased in *cis*). To ensure accuracy of our subsequent analyses, we manually inspected read alignments for all 190 individuals with more than one rare heterozygous *RNU2-2* variant. We identified 15 phase switch errors[12,20], including seven individuals with variants in *trans* incorrectly phased in *cis*, and eight with variants in *cis* incorrectly phased in *trans*. After accounting for these errors, we identified 33 homozygous and 75 confident compound heterozygous genotypes in 100kGP (Fig. 1c and Extended Data Fig. 2).

We next searched for additional individuals in 100kGP not included in the statistically phased aggregated variant dataset, totaling 10,564 additional samples: 2,481 aligned to GRCh38 and 8,083 aligned to GRCh37. Given that the *RNU2-1* repeat array is not annotated in GRCh37 (ref. 21), reads originating from *RNU2-1* are often mapped as low-quality *RNU2-2* reads. This process reduces the alternate allele fraction of true *RNU2-2* variants and increases the rate of false-negative variant calls. We mapped *RNU2-2* reads from the GRCh37-mapped samples to GRCh38. For individuals with more than one heterozygous *RNU2-2* variant, we confirmed variant phasing by visual inspection of reads in the Integrative Genomics Viewer (IGV)[22]. We identified a further 14 individuals with biallelic *RNU2-2* variants (four homozygous, ten compound heterozygous; Fig. 1c and Extended Data Fig. 3).

In total, we identified 122 individuals with rare biallelic *RNU2-2* variants from 100kGP. These included 37 individuals from 30 families with 25 distinct homozygous variants (17 with unsolved NDDs

and 20 controls) and 85 individuals from 82 families with 80 distinct compound heterozygous genotypes (23 with unsolved NDDs and 62 controls) (Fig. 1c and Supplementary Table 3).

### The distribution of candidate variants is distinct from controls

*RNU2-2* contains four evolutionarily conserved 5′ modules (stem I, the branchpoint recognition sequence, stem II and the Sm binding site) and two less conserved 3′ modules (stem III and stem IV)[23–26]. Visualization of biallelic variants in the U2-2 primary and secondary structure suggested a clustering of deleterious variants in the 5′ end of the transcript (Fig. 2 and Extended Data Fig. 4a,b). We compared the distributions of biallelic *RNU2-2* variants in the unsolved NDD cohort versus 100kGP controls and UK Biobank (UKB) controls within this 5′ constrained region, which we conservatively extend to n.67 to coincide with the junction of stem loop IIa and stem loop IIb (Extended Data Fig. 5). Homozygous variants were scarce in all three cohorts, and we found no significant difference in the number of unique homozygous variants within, or outside of, n.1–n.67 (NDD vs 100kGP controls: OR = 1.56, 95% CI = 0.0860–28.1, two-tailed Fisher's exact test $P$ = 1; NDD vs UKB, OR = 2.78, 95% CI = 0.157–49.2, $P$ = 0.484) (Supplementary Table 4). However, for compound heterozygous genotypes, we observed a strong enrichment of genotypes with at least one variant within n.1–n.67 in individuals with unsolved NDD versus both 100kGP controls (OR = 34.0, 95% CI = 6.81–170, two-tailed Fisher's exact test $P$ = 1.07 × 10⁻⁷) and 200,011 individuals in UKB for whom statistically phased genome sequencing data were available[12] (OR = 153, 95% CI = 19.8–1,180, $P$ = 1.90 × 10⁻¹⁰). In other recessive RNU-opathies, variants in the Sm binding site are known to be pathogenic[15,27]. However, we did not detect statistical enrichment of homozygous or compound heterozygous genotypes in the Sm site (n.97–n.107) in individuals with unsolved NDD versus controls (two-tailed Fisher's exact test $P$ > 0.0565) (Supplementary Table 4).

### Validation across multiple cohorts confirms *RNU2-2* as a novel recessive disease gene

To prioritize probable disease-causing variants in our unsolved NDD cohort, we filtered out any individuals with homozygous or compound heterozygous variants that were also observed in population databases (gnomADv4 (ref. 28), UKB[29] or All of Us[30]) or in our 100kGP controls in the same combination. Specifically, compound heterozygous genotypes were filtered from the unsolved NDD cohort if the same combination of variants, that is, the same compound heterozygous genotype, was observed in any control cohort. Ultimately, we prioritized 31 rare biallelic genotypes in *RNU2-2* (ten homozygous and 21 compound heterozygous) in 38 individuals from 31 families with unsolved NDD. These formed our 'discovery cohort' in 100kGP (Fig. 1c and Supplementary Table 5).

To validate our findings, we searched for candidate biallelic *RNU2-2* variants in additional rare conditions databases. Specifically, we looked for biallelic variants absent in the same combination from UKB, gnomADv4 or 100kGP controls. For compound heterozygous variants, guided by our observations in the discovery cohort, we used a stringent approach to reduce the identification of false positives and included only those genotypes in which at least one variant was in the 5′ constrained region (n.1–n.67) or the Sm site (n.97–n.107). Using these criteria, we identified ten additional individuals in the National Health Service (NHS) Genomic Medicine Service (GMS) ($n$ = 29,872 genomes from 15,889 families with rare conditions), two in Solve-RD[31] ($n$ = 334 genomes), two in UDN-Aus[32] ($n$ = 249 genomes from 94 families), 13 through the national LoqusDB[33] database at Karolinska University Hospital, Sweden ($n$ = 29,782 clinical genomes from Stockholm Region and 141 genomes for UDN Sweden), four in the South Korean Undiagnosed Diseases database ($n$ = 1,089 whole genome sequencing probands) and 14 cases were identified from the Lifer Omics Database in Saudi Arabia ($n$ = 6,657 genomes). Notably, although we did not filter by phenotype at this stage, all 45 individuals had unsolved NDD

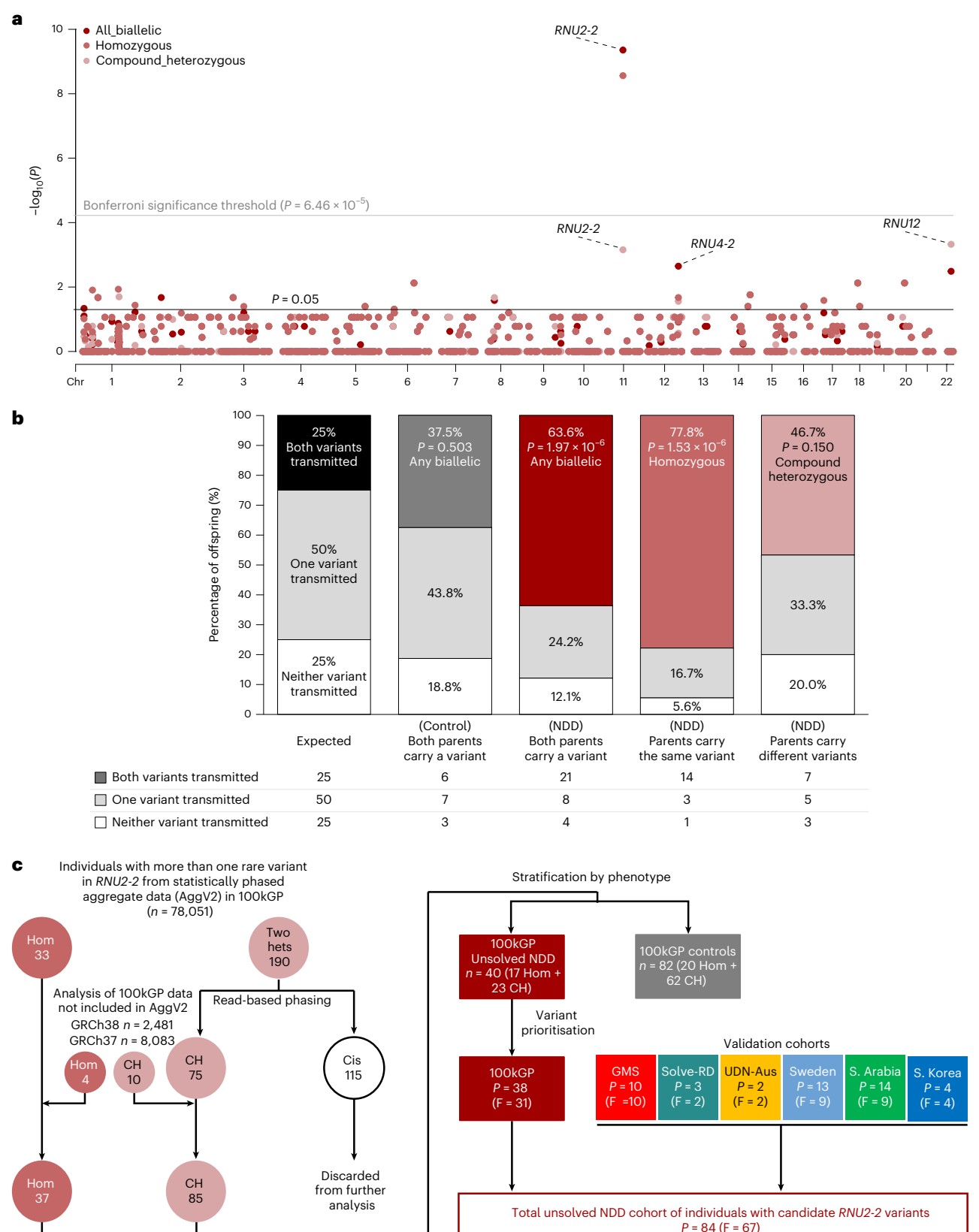

**Fig. 1 | Biallelic variants in *RNU2-2* are enriched and over-transmitted in individuals with unsolved NDD. a**, Manhattan plot showing biallelic variant enrichment for all snRNAs (*n* = 1,901). *P* values from two-sided Fisher's exact tests are shown for homozygous variants, compound heterozygous variants and all biallelic variants. The Bonferroni significance threshold (*P* = 6.46 × 10⁻⁵ for 774 tests at α = 0.05) is indicated with a gray line. **b**, Stacked bar plot showing transmission of heterozygous variants in *RNU2-2* from parents to offspring. Only trios in which both parents are carriers of heterozygous variants in *RNU2-2* are shown (*n* = 49). **c**, Flowchart describing the variant identification and filtering strategy. Hom, homozygous; CH, compound heterozygous; Cis, variants in *cis*. The data underlying the Manhattan plot are provided as Source Data.

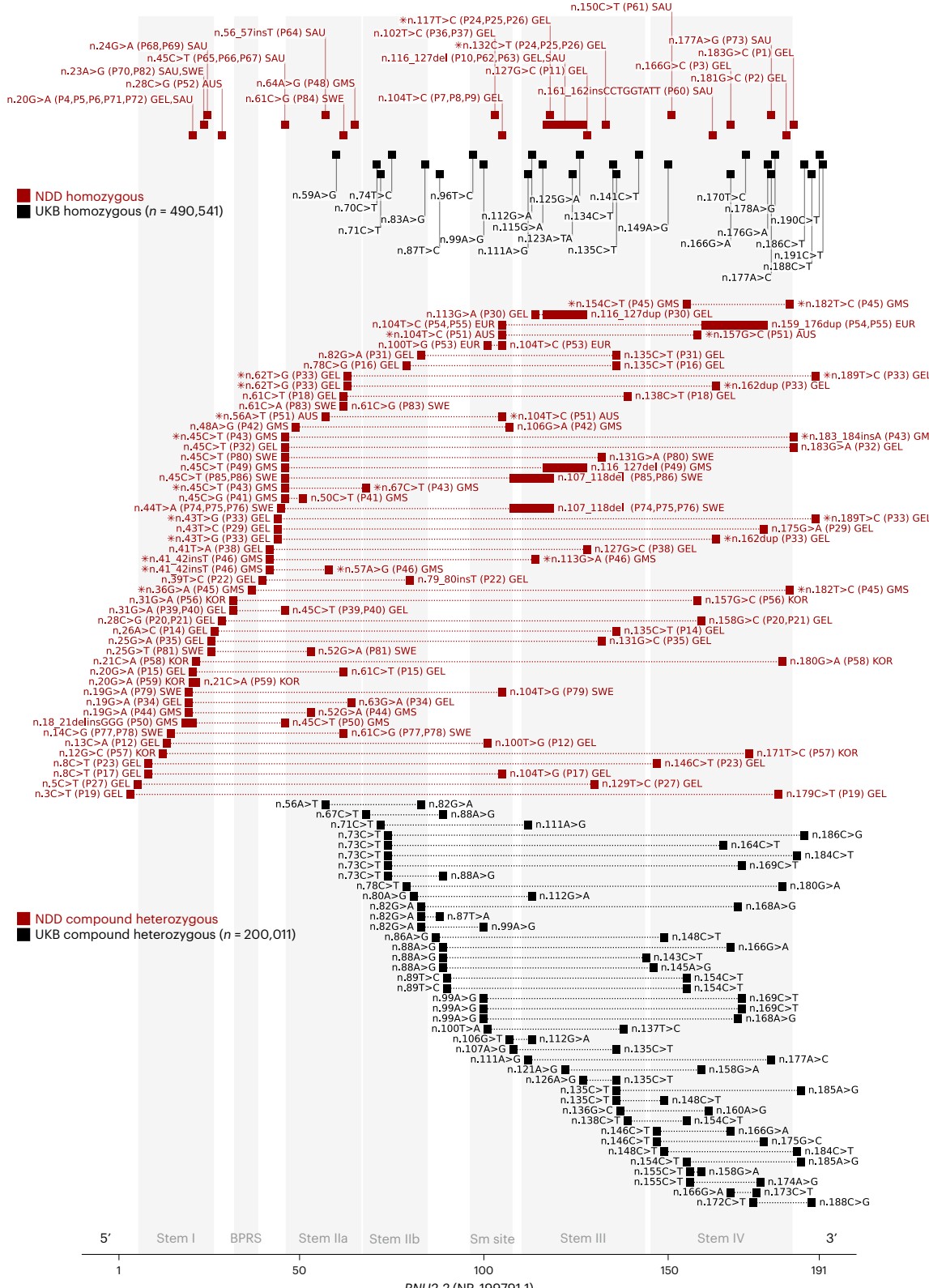

**Fig. 2 | Biallelic variant distributions in *RNU2-2* (NR_199791.1) among unsolved NDD cases and controls.** Transcript diagram showing biallelic variants in *RNU2-2*. Red squares depict candidate biallelic variants in individuals with unsolved NDD from the discovery cohort (GEL), GMS, Solve-RD (EUR), UDN-Aus (AUS), Saudi Arabian (SAU), South Korean (KOR) and Swedish (SWE) cohorts. Gray squares show biallelic variants in UKB controls. Closed squares show homozygous variants. Open squares show compound heterozygous variants. For candidate variants depicted in red, the numbers in parentheses show the participant identifier used in this study. Individuals carrying more than two variants are marked with an asterisk. In such cases, all compound heterozygous variant combinations are shown, except for individuals with more than one homozygous variant, for whom only homozygous variants are shown. Indels are plotted at their most 5′ coordinate. Genotypes that contain variants extending past the start or end coordinates of *RNU2-2* are not shown. The extent of the branchpoint recognition sequence (BPRS), Sm site, stem loops I, IIa, IIb, III and IV are highlighted in gray. The boundary of the 5′ constrained region at n.67 is shown by whitespace between stem loop IIa and stem loop IIb.

(Fig. 1c). Finally, we identified three additional affected siblings across three cohorts through segregation analysis.

In total, combining individuals from our discovery and validation cohorts, we identified 84 individuals with NDD phenotypes from 67 families with candidate biallelic *RNU2-2* variants (Fig. 2, Extended Data Fig. 5, Supplementary Fig. 1 and Supplementary Table 5), only one of whom (P56 in Supplementary Table 5) had another molecular explanation identified by prior comprehensive genomic testing.

## The recessive *RNU2-2* syndrome is a DEE

We examined the phenotypic similarity of 31 unrelated individuals with candidate biallelic *RNU2-2* variants from our discovery cohort. We excluded seven siblings from this analysis to prevent confounding owing to relatedness. For these 31 individuals, we compared the Human Phenotype Ontology (HPO) terms documented at recruitment with 1,000 permutations of 31 randomly selected unrelated individuals with NDD in 100kGP. HPO terms in the 31 individuals with candidate *RNU2-2* variants were significantly more homogenous than expected by chance (two-sided Monte Carlo $P = 0.012$; Fig. 3a).

To identify the clinical features underlying this phenotypic similarity, we performed HPO term enrichment analysis for these 31 unrelated individuals versus all other unrelated participants with NDD in 100kGP ($n = 10,157$). Significantly enriched HPO terms were consistent with a DEE and included 'generalized seizures', 'encephalopathy', 'intellectual disability' and 'delayed speech and language development' (Fig. 3b and Supplementary Table 6).

Next, we performed a detailed clinical characterization of the recessive *RNU2-2* syndrome. We gathered detailed phenotypic information and clinical histories from 34 individuals (27 families) with candidate biallelic *RNU2-2* variants. Of these individuals, 16 were male and 18 were female, and their ages ranged between 5 and 34 years (Supplementary Table 7; case reports in Supplementary Note). All individuals displayed developmental delay and intellectual disability. Most individuals had motor delay (32 out of 34, 94.1%) and seizures (32 out of 34, 94.1%). Seizure onset occurred in the first year of life for 61.8% (21 out of 34). 73.5% (25 out of 34) of individuals were non-verbal and 41.2% (14 out of 34) were unable to walk. Seizure semiology was varied with generalized tonic-clonic seizures in 61.8% (21 out of 34), myoclonic jerks in 38.2% (13 out of 34), absence seizures in 32.4% (11 out of 34) and infantile spasm in 8% (three out of 34). Encephalopathy was used as a clinical descriptor in 23.5% (eight out of 34). Movement disorders were seen in 35.3% (12 out of 34). Bruxism was noted in 17.6% (six out of 34). Gastrostomy feeding was required in 47%. Electroencephalograms were abnormal in 53.8% (14 out of 26), with epileptiform spike activity in 50% (13 out of 26) and background slowing indicative of encephalopathy in 26.9% (7 out of 26). Magnetic resonance imaging scans were abnormal in 46% (13 out of 28), with cerebral atrophy being most common (nine out of 28), followed by enlarged extra-axial cerebrospinal fluid spaces in 14% (four out of 28) (Fig. 3c). Facial dysmorphism was seen in most individuals (20 out of 34) (Fig. 3d).

In summary, we demonstrate striking phenotypic convergence in a reverse-phenotyped[34] cohort with candidate biallelic *RNU2-2* variants and show that the recessive *RNU2-2* disorder is a DEE.

## Candidate biallelic variants are associated with reduced U2-2 abundance and a decreased U2-2:U2-1 ratio

Recessive inheritance of the disorder described here suggests loss-of-function as the probable pathomechanism, which can occur by several means, including reduced transcript stability[35–39]. We examined rRNA-depleted RNA sequencing data from blood in 100kGP and performed expression outlier analysis in 5,412 samples using OUTRIDER[40]. Nine individuals in this cohort had candidate biallelic *RNU2-2* variants, and eight of these individuals had significantly lower *RNU2-2* expression in OUTRIDER (false discovery rate

(FDR) $P < 0.05$) (Supplementary Table 8), whereas only six individuals in the remainder of the cohort ($n = 5,403$) were significant expression outliers for *RNU2-2*. Overall, *RNU2-2* expression outliers were highly over-represented among individuals with candidate *RNU2-2* variants (OR = 7,372, 95% CI = 795–68,400, $P = 4.68 \times 10^{-15}$), suggesting reduced U2-2 expression or reduced transcript stability as a potential mechanism in most individuals with this disorder.

Interestingly, seven out of eight individuals with outlier *RNU2-2* expression were also found to be an expression outlier for *WDR74* (Supplementary Table 8). *RNU2-2* is located within the 5' untranslated region of *WDR74* on chromosome 11 (Extended Data Fig. 6a)[21]. We observed no significant difference in the number of RNA sequencing reads mapping to exclusively coding exons of *WDR74* between cases and controls (mean reads per million (RPM) of 96.8 and 85.6, respectively; two-tailed Mann–Whitney *U*-test $P = 0.151$) (Extended Data Fig. 6b,c). This implies that the apparent reduced *WDR74* expression in individuals with candidate *RNU2-2* variants is a technical artefact.

Next, we explored the transcriptional relationship of U2-1 and U2-2 in cases and controls in these transcriptomic data. First, we counted reads mapping to *RNU2-2* in the nine individuals with candidate biallelic variants and in an expanded set of control samples ($n = 5,443$). Consistent with our OUTRIDER results, *RNU2-2* expression was significantly lower for individuals with biallelic variants than controls (mean RPM of 102 vs 757; two-tailed Mann–Whitney *U*-test $P = 1.53 \times 10^{-6}$) (Fig. 4a and Supplementary Table 9). Next, we found that expression of U2-1 and U2-2 is highly correlated (Pearson's $R = 0.83$, $P < 2.2 \times 10^{-16}$; Fig. 4b), consistent with previous results[11]. We did not detect any significant OUTRIDER expression outliers for *RNU2-1* among any individuals in the original cohort of 5,412 samples. Strikingly, all eight *RNU2-2* OUTRIDER outliers among the cases had a significantly depleted ratio of U2-2:U2-1 expression (range, 0.352–0.688). No other individuals, including the six *RNU2-2* outliers from the control cohort, had a U2-2:U2-1 ratio in this range ($n = 5,448$; range, 0.76–1.45). The mean *RNU2-2* ratio was significantly lower for the participants with candidate biallelic variants than controls (0.56 vs 1.04, two-sided Mann–Whitney *U*-test $P = 4.4 \times 10^{-7}$) (Fig. 4c). These data suggest that the U2-2:U2-1 ratio is a more specific marker for the recessive *RNU2-2* disorder than U2-2 expression alone.

## Dominant and recessive *RNU2-2* syndromes are distinct

We next set out to compare the dominant[10,11] and recessive *RNU2-2* disorders. The dominant *RNU2-2* disorder is caused by a limited repertoire of single nucleotide variants at the 5' end of *RNU2-2* (Extended Data Fig. 5), including highly recurrent variants at n.4 G > A and n.35 A > G[10,11]. Candidate biallelic variants in the recessive disorder are more widely distributed throughout *RNU2-2*, notwithstanding an enrichment of compound heterozygous alleles at the 5' end of the gene (Extended Data Fig. 5). Overall, the genotypic spectrum of the recessive disorder appears much broader than the dominant condition.

Next, we compared these disorders at the RNA level. Having shown that candidate biallelic variants are associated with reduced U2-2 abundance and a reduced U2-2:U2-1 ratio, we found that mean U2-2 abundance in individuals with the dominant condition was no different from controls (mean RPM of 745 vs. 955; two-tailed Mann–Whitney *U*-test $P = 0.924$) (Fig. 4a and Supplementary Table 10). Similarly, the U2-2:U2-1 ratio in these heterozygous participants did not differ significantly from controls (range, 0.88–1.06 vs 0.76–1.08; two-tailed Mann–Whitney *U*-test $P = 0.07$) (Fig. 4c). These data suggest that U2-2 transcript depletion is probably a specific feature of the recessive disorder.

From 5,546 participants with rare conditions for whom transcriptomic data were available, we identified five individuals with pathogenic heterozygous variants and nine individuals with candidate biallelic *RNU2-2* variants. We compared these cases with a subset of 301 controls with non-NDD phenotypes who were under 18 years

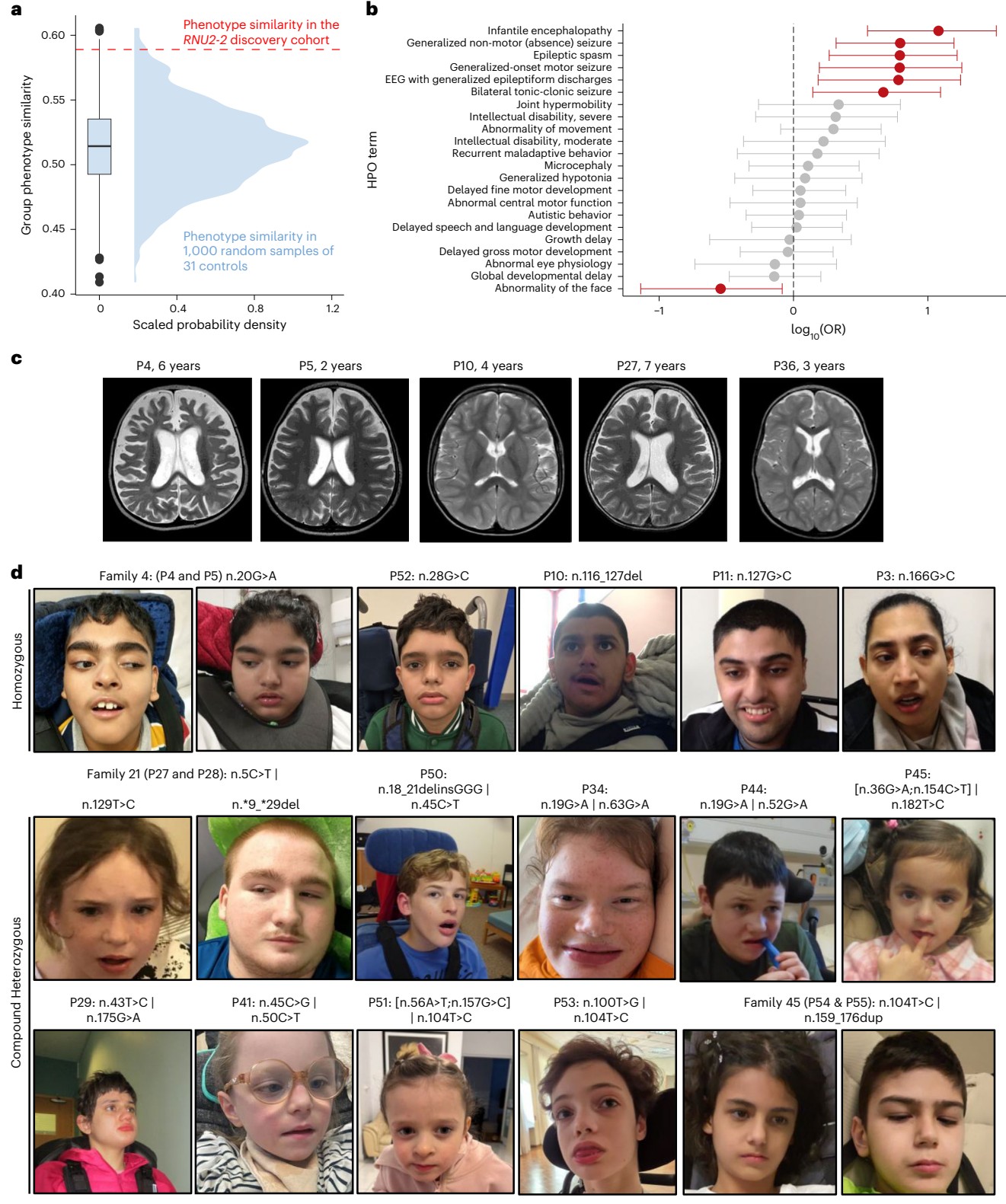

**Fig. 3 | The recessive *RNU2-2* syndrome is a DEE. a**, Kernel density estimate and boxplot showing the group phenotypic homogeneity of 1,000 permutations of 31 randomly selected participants with NDDs in 100kGP versus 31 individuals with candidate biallelic variants in *RNU2-2* (indicated by dashed red line). The boxplot shows the median, quartiles and ±1.5 times the interquartile range of the data. Outliers are shown by filled black circles. **b**, Relative frequency of HPO terms in unrelated individuals with rare biallelic variants in *RNU2-2* versus unrelated 100kGP participants with NDD. HPO terms that differ in frequency between the cohorts are shown in red (two-sided Fisher's exact tests with Benjamini–Hochberg

FDR correction for 71 tests at α = 0.05, *P* < 0.000704). Center points represent the odds ratio; error bars, 95% confidence intervals. The raw data are provided in Supplementary Table 7. **c**, T2-weighted axial brain MRI images of individuals with candidate variants in *RNU2-2* showing enlarged CSF spaces and generalized cerebral atrophy (P4, P5, P27 and P36). Prominent cerebral atrophy in the temporal regions is shown for P10. **d**, Facial photographs of individuals with candidate biallelic variants in *RNU2-2*. Participant numbers and variant details in HGVS nomenclature are shown. Informed consent was obtained from all families for the publication of facial images. Data underlying the plots are provided as Source Data.

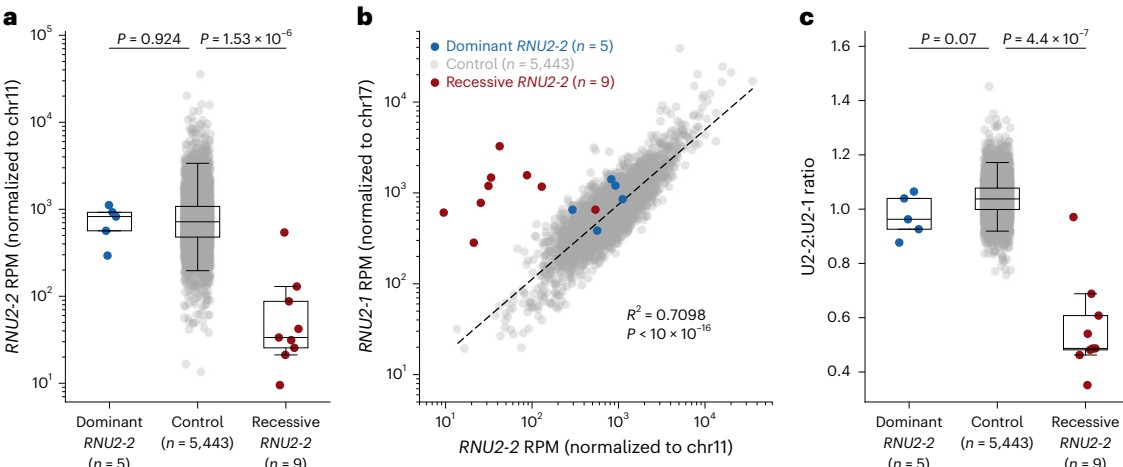

**Fig. 4 | U2-2 and U2-1 transcript abundance in carriers of candidate variants in *RNU2-2* versus controls. a**, Boxplots showing *RNU2-2* RPM, normalized to the sum of reads arising from chromosome 11. **b**, Scatterplot comparing relative expression of *RNU2-2* and *RNU2-1*. The dashed black line shows the ordinary least squares regression line for controls. The $R^2$ and two-sided *P* value for this correlation are shown. *RNU2-2* reads were normalized to the sum of reads arising from chromosome 11, while *RNU2-1* reads were normalized to the sum of

reads arising from chromosome 17. **c**, Boxplots showing the ratio of U2-2:U2-1 (RPM). Individuals with candidate biallelic variants in *RNU2-2* are shown in red. Individuals with pathogenic heterozygous variants in *RNU2-2* are shown in blue. For **a** and **c**, *P* values for two-tailed Mann–Whitney *U*-tests are shown. Box and whisker plots show the median, quartiles and ±1.5 times the interquartile range of the data. Data underlying the plots are provided as Source Data.

of age. We did not observe a significant difference in the number of FDR-significant (FDR $P < 0.10$) splicing outliers between groups (Extended Data Fig. 7a). Broadening our analysis to nominally significant ($P < 0.05$) FRASER2 outliers showed that individuals with the dominant disorder have a significantly greater number of nominally significant splicing outliers in blood than controls (mean splicing outliers, 7,072 vs 5,267; two-tailed Mann–Whitney *U*-test $P = 0.0112$), whereas individuals with candidate biallelic variants did not (mean, 5,722 vs 5,266; $P = 0.695$) (Fig. 5a). Given the prominence of seizures in the recessive disorder, we further subset our analysis to nominally significant splicing outliers in known monogenic epilepsy genes (PanelApp Australia v.1.178 Genetic Epilepsy panel). Again, we found that individuals with the dominant disorder had a significantly greater number of splicing outliers in these genes than controls (mean, 482 vs 350; two-sided Mann–Whitney *U*-test $P = 0.00752$), but we found no such difference in candidate biallelic cases (mean, 358 vs 350; $P = 0.825$) (Extended Data Fig. 7b). We next looked for aberrant splicing events shared by individuals with the dominant or recessive disorders. We compared the number of identical, nominally significant aberrant splicing events shared by individuals with candidate *RNU2-2* variants versus 1,000 permutations of an equal number of randomly selected controls. Individuals with the dominant disorder shared a greater number of aberrant splicing events at annotated splice junctions than expected by chance ($n = 1,930$; two-sided percentile bootstrap $P = 0.002$, Bonferroni $P = 0.016$) (Extended Data Fig. 8). No such signal was observed for participants with candidate biallelic variants (Extended Data Fig. 9). In summary, we were able to detect an aberrant splicing signal in blood for individuals with the dominant disorder but not with the recessive disorder.

Finally, we compared the clinical phenotypes of the dominant and recessive conditions using detailed clinical information from 26 individuals with the recessive *RNU2-2* NDD as well as published clinical data for 21 individuals with the dominant *RNU2-2* disorder[10,11]. We limited participants with the dominant condition to those carrying the highly recurrent n.4 G > A, n.35 A > G and n.35 A > C variants, as only these variants reach the threshold for a '(Likely) Pathogenic' classification by current ACMG criteria[41]. Principal component analysis of 45 clinical features showed limited separation of the dominant and recessive *RNU2-2* conditions (Fig. 5b and Supplementary Fig. 2).

We therefore compared the frequency of individual phenotype terms between the recessive and dominant disorders. The most strongly enriched clinical features for the recessive disorder included 'spasticity' (OR = 18.26, 90% CI = 1.59–208, two-tailed Fisher's exact test FDR $P = 0.038$) and 'seizures in first year' (OR = 5.60, 90% CI = 1.99–15.7, FDR $P = 0.039$; Fig. 5c and Supplementary Table 11). The most strongly enriched phenotypes for the dominant disorder included 'stereotyped hand movements' (OR = 0.064, 90% CI = 0.018–0.22, FDR $P = 4.5 \times 10^{-4}$) and dysmorphic facial features (OR = 0.031, 90% CI = 0.002–0.22, FDR $P = 0.029$; Fig. 5c).

Taken together, these data suggest that the dominant and recessive *RNU2-2* disorders are genetically, molecularly and clinically distinct.

### Biallelic *RNU2-2* variants cause the most common recessive NDD in 100kGP

Finally, we estimated the frequency of the recessive *RNU2-2* disorder compared to other recessive NDDs. We compared the number of individuals with candidate biallelic *RNU2-2* variants versus the number of individuals with a confirmed recessive NDD using exit questionnaire data from 10,157 individuals with NDD in 100kGP. The recessive *RNU2-2* disorder is by far the most frequent recessive condition in this cohort ($n = 31$), observed in over three times as many individuals as *VPS13B* (Cohen syndrome, MIM 216550; $n = 9$), the next most frequent diagnosis (Fig. 6a). Expanding our analysis to all NDD genes with 'green' review status in the intellectual disability panel (R29) in PanelApp[42], we find that the recessive *RNU2-2* disorder is the second-most frequent diagnosis after *RNU4-2*-related ReNU syndrome ($n = 59$) and the only recessive disorder among the top 20 most frequent diagnoses in this cohort (Extended Data Fig. 10).

We previously identified 11 individuals with the dominant *RNU2-2* syndrome in 100kGP[10]. Now, the discovery of 38 individuals with the recessive *RNU2-2* syndrome brings the total number of individuals in this cohort with an *RNU2-2*-related diagnosis to 49 out of 7,968 (0.61%).

The recently discovered RNU-opathies (dominant and recessive *RNU4-2* (refs. 14,15,43,44), dominant and recessive *RNU2-2* (refs. 10,11) and dominant *RNU5B-1* syndromes[10]) account for 118 out of 7,968 (1.48%) individuals with previously unsolved NDDs in 100kGP (Fig. 6b). This finding is particularly remarkable given

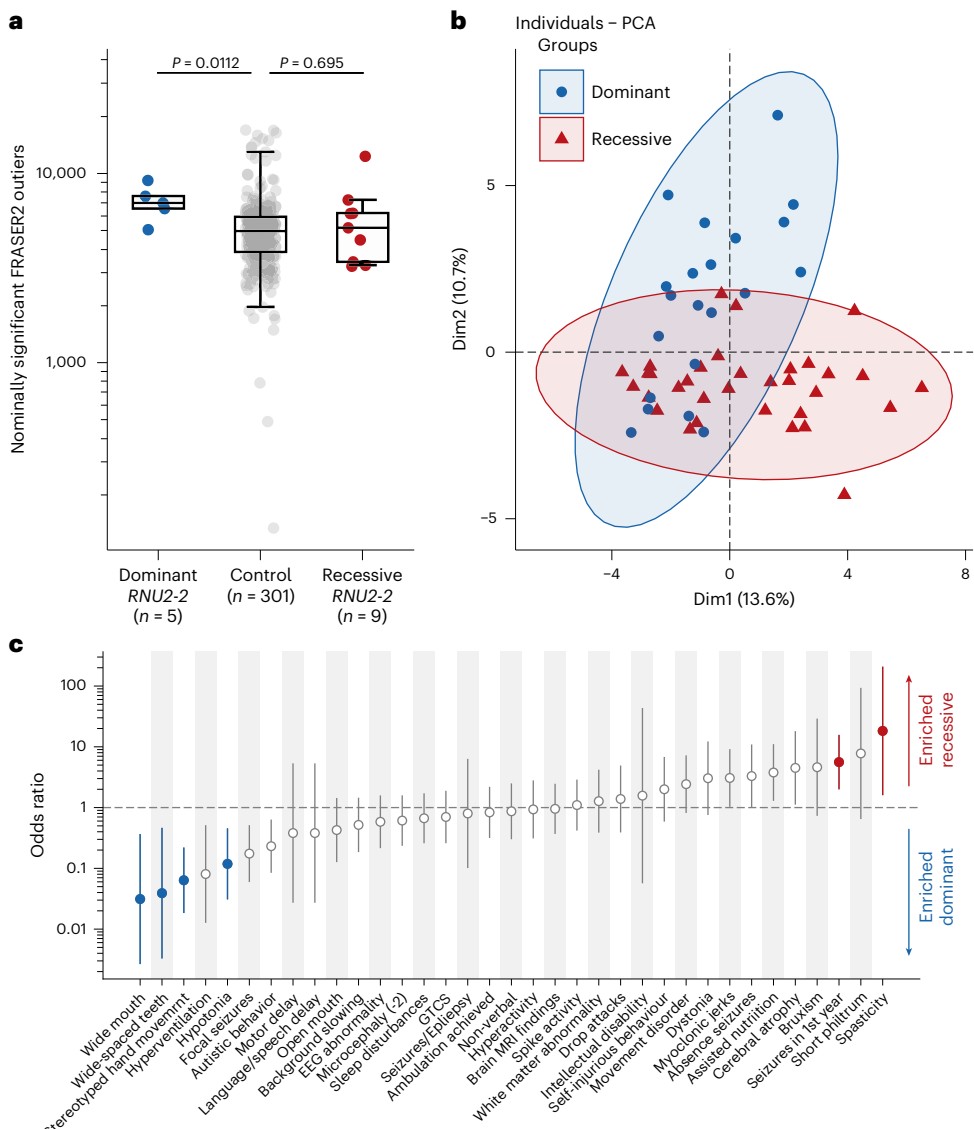

**Fig. 5 | Molecular and phenotypic distinction between the recessive and dominant *RNU2-2* disorders. a**, Nominally significant FRASER2 outliers (*P* < 0.05) in carriers of candidate variants in *RNU2-2* versus controls. *P* values from two-tailed Mann–Whitney *U*-tests are shown. Box and whisker plots show the median, quartiles and ±1.5 times the interquartile range of the data. **b**, PCA scatterplot of HPO terms in the recessive and dominant *RNU2-2* disorders. The proportion of the variance explained by each principal component is shown in the axis labels. Ellipses were generated using the default parameters of the factoextra R package. **c**, Relative frequency of HPO terms in the recessive and dominant *RNU2-2* disorders. Odds ratios after Haldane–Anscombe correction are shown. Error bars show 90% confidence intervals, centered around the odds ratio represented as a circle. Filled circles show FDR-significant *P* values from two-sided Fisher's exact tests (α = 0.05). Data underlying the plots are provided as Source Data.

that these genes have a combined genomic footprint of 448 bp (that is, one in $1.35 \times 10^7$ bp of GRCh38).

In summary, biallelic *RNU2-2* variants are a highly frequent unrecognized cause of NDD with seizures.

## Discussion

Here, we describe an NDD caused by rare biallelic *RNU2-2* variants, define the genetic architecture of the disorder, characterize its clinical presentation, show that most candidate variants probably result in transcript depletion, identify an RNA-based potential diagnostic biomarker for the condition and find that the recessive *RNU2-2*-related syndrome is the most common recessive NDD in 100kGP. Our conclusions are supported by other recent independent studies[45,46], one of which also uses the 100kGP cohort[46].

*RNU2-2* is a single-exon non-coding gene of only 191 bp. Given this compact size, it is remarkable that variants in this gene are

responsible for such a frequent recessive disorder. Notably, our ascertainment criteria were stringent, and it is possible that some pathogenic variant combinations could have been filtered out.

Among our candidate cases in 100kGP, compound heterozygous genotypes were more frequent than homozygous genotypes, and we did not identify a high number of consanguineous families. The distribution of compound heterozygous variants among our cases suggests that specific variant combinations are important to the etiology of the disorder. We speculate that a combination of two strongly deleterious *RNU2-2* alleles may be non-viable, while two mildly deleterious alleles may not cause a clinically obvious phenotype, and the recessive *RNU2-2*-related syndrome occurs only within a specific window of residual U2-2 function. We note that homozygous variants are scarce in the 5′ end of *RNU2-2* in both disease and control populations, whereas this region is enriched for compound heterozygous variants in individuals with NDD, suggesting that some 5′ homozygous variants

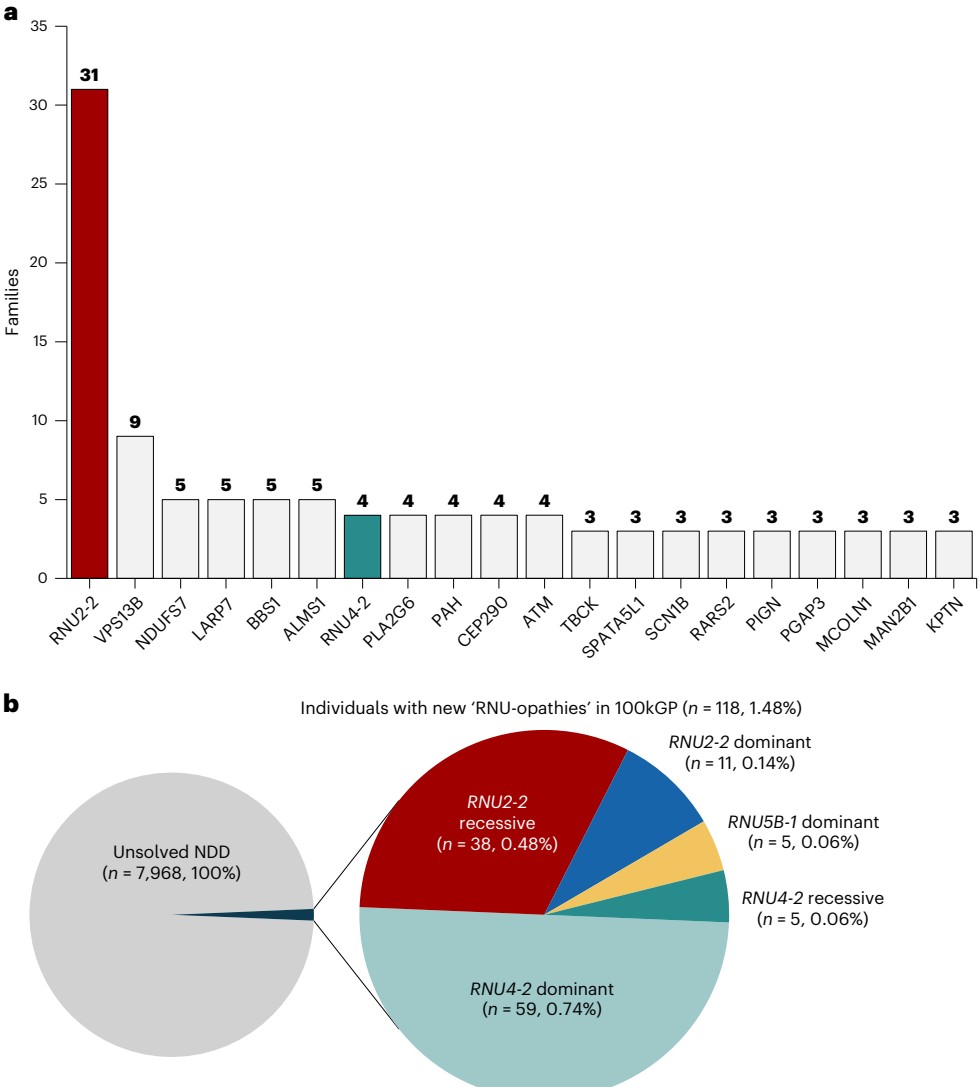

**Fig. 6 | The frequency of the recessive *RNU2-2* syndrome and other 'RNU-opathies' in 100kGP. a**, Bar plot showing the 20 most frequent recessive diagnoses in the rare disease arm of 100kGP. The number of families 'solved' by biallelic variants in each gene is shown. Counts for the *RNU2-2* and *RNU4-2* disorders were based on genotyping (see Methods). Counts for all other genes were taken from the exit questionnaire data. **b**, Pie chart showing the number of individuals with unsolved NDD in 100kGP explained by pathogenic variants in the recently described 'RNU-opathy' genes. Note that **a** shows counts of families, whereas **b** shows counts of individuals.

could be lethal. Such a 'Goldilocks' phenomenon has precedent in other small non-coding RNA genes associated with recessive conditions, including *RNU7-1* (ref. 47) and *SNORD118* (ref. 27).

The broad genotypic spectrum of this recessive disorder presents an obstacle to variant interpretation in the clinical setting. We provide strong statistical and clinical evidence for *RNU2-2* as a novel recessive disorder gene, whereas many of our candidate variants will be classed as variants of uncertain significance according to clinical variant interpretation guidance. The interpretation of non-coding variants is known to be especially challenging[48]. Future studies that leverage larger cohorts of affected individuals and variant-specific functional data using approaches such as saturation genome editing[14] can facilitate clinical variant interpretation.

We show that the recessive *RNU2-2*-related syndrome is a DEE characterized by severe to profound global developmental delay, intellectual disability, early-onset seizures of varying semiology, movement disorders, requirement of gastrostomy feeding, abnormal electroencephalograms and evidence of cerebral atrophy on neuroimaging. More detailed natural history studies will be essential to investigate the range and frequency of clinical features and comorbidities caused by this disorder.

Consistent with a recessive mode of inheritance, we show that candidate biallelic *RNU2-2* variants are associated with U2-2 transcript depletion, indicative of a loss-of-function mechanism. We reason that the transcript depletion we observe may be secondary to reduced transcript stability. Understanding the molecular basis of this disorder will be critical for the development of targeted therapies.

Importantly, we find that a reduced U2-2:U2-1 ratio is a specific marker for the recessive disorder. This finding is important because we observed some controls with significantly low U2-2 expression but a preserved U2-2:U2-1 ratio. These paralogous genes differ at only nine nucleotides (Supplementary Fig. 3), four of which are weakly conserved positions adjacent to the Sm site[10]. It is therefore plausible that U2-1 may partially compensate for under-expression of U2-2 (ref. 49). This compensatory effect may be especially true in blood, in which the expression of U2-1 relative to U2-2 is greater than in brain or retinal tissue[10] and the expression of both genes is highly correlated. A reduced U2-2:U2-1 ratio in blood implies either stable U2-1 expression

independent of U2-2 levels or compensatory over-expression of U2-1 in response to U2-2 depletion. The technical limits of short-read sequencing for repetitive regions make a more in-depth analysis of the *RNU2-1* locus in this current cohort challenging.

Deleterious variants in a core component of the spliceosome may be expected to disrupt splicing. We observed a measurable increase in aberrant splicing events in blood from individuals with the dominant *RNU2-2* disorder but not in the recessive disorder. Further RNA studies are warranted to clarify the splicing defect of the recessive *RNU2-2* disorder, using other sequencing approaches, patient-derived cell lines, clinically relevant tissues or in vitro and in vivo model systems, including animal models. Of note, the *RNU2-2* gene is disrupted in rodent models (Supplementary Fig. 4).

Although our findings will need replication in other populations, we found that *RNU2-2* was the most common recessive NDD in the 100kGP. The discovery of a highly prevalent severe recessive disorder is especially important because accurate carrier testing owing to family history or through extended carrier screening programs can inform reproductive decisions, including prenatal testing. Several factors may explain why the genetic basis of such a prevalent recessive disorder has remained elusive until now. These include the absence of *RNU2-2* in exome capture kits, previous annotation of *RNU2-2* as a pseudogene and confounding of accurate detection of *RNU2-2* variants owing to the absence of the *RNU2-1* locus from the GRCh37 assembly. Finally, the high density of variants in the gene, the relative paucity[50] of founder variants and the preponderance of compound heterozygous genotypes in this condition reduce the efficacy of traditional recessive disease gene discovery approaches. Epidemiological studies to determine the incidence, prevalence and carrier rates for the recessive *RNU2-2* syndrome will be required in the future.

In summary, we describe a highly prevalent NDD and DEE caused by biallelic *RNU2-2* variants, which will enable diagnosis for thousands of individuals with unsolved NDDs worldwide and catalyze future research into the biology, epidemiology and treatment of this condition.

## Online content

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

**Adam Jackson** [1,2,53] ✉, **Alexander J. M. Blakes** [1,2,53], **Bader Alhaddad**[3], **Olivia J. Henry** [4], **Angelica M. Delgado-Vega** [4,5], **Elizabeth Wall**[6], **Ola Abdelhadi**[1,2], **Shakti Agrawal**[7], **Khadijah Bakur**[3], **Edward Blair**[8], **Angela F. Brady**[9,10], **Helen Brittain**[6], **Kate E. Chandler**[1,2], **Natasha Clarke**[11], **Miriana Danelli**[12], **Nicholas Drinkall**[13], **Irene Duba** [4,14], **Frances Elmslie** [11], **Jamie Ellingford** [1,15], **Lisa J. Ewans**[16,17,18], **Andrew P. Fennell** [19,20], **Gabriella Gazdagh**[21,22], **Simon P. Heller** [23], **Anna Hammarsjö**[4,5], **Kristina Karrman**[24], **Usha Kini**[8], **Nicole Lesko**[4,14], **Anna Lindstrand**[4,5], **Rebecca Macintosh** [16,18], **Sahar Mansour**[11,25], **Lara Menzies**[26,27], **Kay Metcalfe**[1,2], **Alison Milhench**[28], **Lina Nashef**[23], **Raymond T. O'Keefe** [1], **Nadja Pekkola Pacheco**[4,5], **Elizabeth E. Palmer** [16,18], **Amitav Parida**[7], **Katrina Prescott**[29], **Melody Redman**[29], **Alessandra Renieri** [12,30,31], **Chiara Fallerini**[12,30,31], **Caterina Lo Rizzo**[12], **Rani Sachdev**[16,18], **Cas Simons** [32,33], **Sanjay M. Sisodiya** [34,35], **Helen Stewart**[8], **Tommy Stödberg**[36,37], **Benito Banos-Pinero**[38], **Fulya Taylan** [4], **Huw B. Thomas** [1], **Flavia Tinella**[12], **Samuel Wiafe**[39,40,41], **Anna Wedell**[4,14], **Nicola Whiffin**[42,43,44], **Susan Walker** [15], **Rocio Rius** [32,33,45], **Jong Hee Chae**[46,47], **Ann Nordgren** [4,5,48,49], **Fowzan Alkuraya** [3,50,51], **Jenny Lord** [52] & **Siddharth Banka** [1,2] ✉

[1]Division of Evolution, Infection and Genomics, School of Biological Sciences, Faculty of Biology, Medicine and Health, University of Manchester, Manchester, UK. [2]Manchester Centre for Genomic Medicine, St Mary's Hospital, Manchester University NHS Foundation Trust, Health Innovation Manchester, Manchester, UK. [3]Lifera Omics, Riyadh, Saudi Arabia. [4]Department of Molecular Medicine and Surgery, Karolinska Institutet, Stockholm, Sweden. [5]Clinical Genetics and Genomics, Karolinska University Hospital, Stockholm, Sweden. [6]West Midlands Regional Genetics Service, Birmingham Women's and Children's NHS Foundation Trust, Birmingham, UK. [7]Birmingham Children's Hospital, Birmingham, UK. [8]Oxford Centre for Genomic Medicine, Oxford University Hospitals NHS Foundation Trust, Oxford, UK. [9]North West Thames Regional Genetics Service, London North West Healthcare University NHS Trust, Northwick Park Hospital, London, UK. [10]Imperial College London, London, UK. [11]South West Thames Centre for Genomics,

St George's, Epsom and St Helier University Hospitals and Health Group, London, UK. [12]Genetica Medica, Azienda Ospedaliera Universitaria Senese, Siena, Italy. [13]Central and South Genomic Laboratory Hub, West Midlands Genomics Laboratory, Birmingham Women's and Children's NHS Foundation Trust, Birmingham, UK. [14]Center for Inherited Metabolic Diseases, Karolinska University Hospital, Stockholm, Sweden. [15]Genomics England, London, UK. [16]Centre for Clinical Genetics, Sydney Children's Hospitals Network, Sydney, New South Wales, Australia. [17]Genomics and Inherited Diseases Program, Garvan Institute of Medical Research, Darlinghurst, New South Wales, Australia. [18]Discipline of Paediatrics and Child Health, School of Clinical Medicine, Faculty of Medicine and Health, University of New South Wales, Sydney, New South Wales, Australia. [19]Monash Genetics, Monash Health, Melbourne, Victoria, Australia. [20]Department of Medicine, School of Clinical Sciences, Monash University, Melbourne, Victoria, Australia. [21]Wessex Clinical Genetic Service, University Hospital Southampton, Southampton, UK. [22]Human Development and Health, Faculty of Medicine, University of Southampton, Southampton, UK. [23]Neurology Department, King's College Hospital, Denmark Hill, London, UK. [24]Department of Clinical Genetics, Pathology and Molecular Diagnostics, Skåne University Hospital, Lund, Sweden. [25]Cardiovascular and Genomics Research Institute, City St George's University of London, London, UK. [26]Department of Clinical Genetics, Great Ormond Street Hospital, London, UK. [27]Department of Genomics and Genetic Medicine, University College London, London, UK. [28]Gloucestershire Hospitals NHS Foundation Trust, Cheltenham, UK. [29]Leeds Clinical Genomics Service, Leeds Teaching Hospitals NHS Trust, Leeds, UK. [30]Medical Genetics, University of Siena, Siena, Italy. [31]Med Biotech Hub and Competence Centre, Department of Medical Biotechnologies, University of Siena, Siena, Italy. [32]Centre for Population Genomics, Garvan Institute of Medical Research, UNSW Sydney, Sydney, New South Wales, Australia. [33]Centre for Population Genomics, Murdoch Children's Research Institute, Melbourne, Victoria, Australia. [34]Research Department of Epilepsy, UCL Queen Square Institute of Neurology, London, UK. [35]Chalfont Centre for Epilepsy, Chalfont Saint Peter, UK. [36]Department of Child Neurology, Karolinska University Hospital, Stockholm, Sweden. [37]Department of Women's and Children's Health, Karolinska Institute, Stockholm, Sweden. [38]Oxford Genetics Laboratories, Oxford University Hospitals NHS Foundation Trust, Oxford, UK. [39]Rare Disease Ghana Initiative, Accra, Ghana. [40]Fred N Binkah School of Public Health, University of Health and Allied Sciences, Ho, Ghana. [41]Ga East Municipal Hospital, Ghana Health Services, Accra, Ghana. [42]Big Data Institute, University of Oxford, Oxford, UK. [43]Centre for Human Genetics, University of Oxford, Oxford, UK. [44]Center for Mendelian Genomics, Program in Medical and Population Genetics, Broad Institute of MIT and Harvard, Cambridge, MA, USA. [45]Department of Paediatrics, The University of Melbourne, Melbourne, Victoria, Australia. [46]Department of Genomic Medicine, Seoul National University Hospital, Seoul, Republic of Korea. [47]Department of Pediatrics, Seoul National University College of Medicine, Seoul National University Children's Hospital, Seoul, Republic of Korea. [48]Department of Clinical Genetics and Genomics, Sahlgrenska University Hospital, Gothenburg, Sweden. [49]Department of Laboratory Medicine, Institute of Biomedicine, Sahlgrenska Academy, University of Gothenburg, Gothenburg, Sweden. [50]Department of Translational Genomics (Genomic Medicine Centre of Excellence (GMCoE)), King Faisal Specialist Hospital and Research Center, Riyadh, Saudi Arabia. [51]College of Medicine, Alfaisal University, Riyadh, Saudi Arabia. [52]Sheffield Institute for Translational Neuroscience (SITraN), The University of Sheffield, Sheffield, UK. [53]These authors contributed equally: Adam Jackson, Alexander J. M. Blakes. ✉e-mail: adam.jackson@manchester.ac.uk; siddharth.banka@manchester.ac.uk

## Methods

### Enrichment analysis for biallelic variants in snRNA genes

A list of snRNAs was generated by filtering the GENCODE comprehensive annotations (v.32) for lines containing the string 'gene_type 'snRNA''. Statistically phased sequencing data, aligned to GRCh38, for 78,051 individuals from 100kGP (AggV2) were accessed within the Genomics England Research Environment (GERE). Biallelic variants in snRNA genes were identified within these data using bcftools (v.1.16)[51] and custom scripts. Only PASS variants were included. Variants occurring above 0.001 allele frequency in AggV2 were filtered.

Individuals were defined to have an 'unsolved NDD' if they fulfilled the following criteria: (1) Individuals with any of the following HPO terms: 'Global developmental delay' (HP:0001263), 'Profound global developmental delay' (HP:0012736), 'Severe global developmental delay' (HP:0011344), 'Moderate global developmental delay' (HP:0011343), 'Mild global developmental delay' (HP:0011342), 'Intellectual disability' (HP:0001249), 'Intellectual disability, profound' (HP:0002187), 'Intellectual disability, severe' (HP:0010864), 'Intellectual disability, moderate' (HP:0002342), 'Intellectual disability, mild' (HP:0001256), 'Intellectual disability, progressive' (HP:0006887), 'Intellectual disability, borderline' (HP:0006889), 'Autism' (HP:0000717), 'Autistic behavior' (HP:0000729) or 'Autism with high cognitive abilities' (HP:0000753) as per previous work[43]; or (2) were recruited under disease category 'Intellectual disability'; and (3) 'Case solved' status did not equal 'yes' in the Genomic Medicine exit questionnaire (gmc_exit_questionnaire table in LabKey); and (4) absent from the 'submitted_diagnostic_discovery' table in LabKey.

In total, 10,987 individuals in 100kGP were identified who had NDD as defined above. A total of 7,968 individuals remained unsolved in the v.4 release (preceding the publication of *RNU4-2*, *RNU2-2* and *RNU5B-1* dominant syndromes); 6,762 individuals with unsolved NDD were identified within the AggV2 data. The remaining 71,289 individuals in AggV2 were defined as '100kGP controls'. The number of homozygous and compound heterozygous variants in each snRNA gene was calculated for both cohorts. For each gene with at least one variant ($n$ = 774), ORs for the number of variants in each cohort were calculated. Statistical significance was determined with two-sided Fisher's exact tests with Bonferroni correction at α = 0.05.

### Transmission analysis of rare heterogeneous *RNU2-2* alleles

All 100kGP trios, aligned to GRCh38, were identified from the 'denovo_cohort_information' table in LabKey ($n$ = 12,015 trios). Parent pairs in which both father and mother were heterozygous for a single rare variant in *RNU2-2* (minor allele frequency of <0.001 in AggV2) were identified using custom scripts. The resulting trios were then stratified according to whether parents were heterozygous for the same variant or for different variants, and whether the offspring was a member of our 'unsolved NDD' or '100kGP control' cohorts.

The *RNU2-2* genotype of the offspring in each trio was determined. The number of times that either both, neither or one allele was transmitted to the offspring was counted. A chi-squared goodness-of-fit test was performed for the observed number of transmissions against the expected number according to Mendelian ratios.

### Phase switching error detection

All individuals in AggV2 with two or more rare variants in *RNU2-2* were identified using bcftools (v.1.16)[52]. Any variants that occurred in AggV2 at allele frequency greater than 0.001 were removed. IGV[22] was used to generate screenshots of the entire *RNU2-2* sequence for all individuals with two or more rare variants. These were inspected manually and were either deemed in *trans* if the variants occurred on mutually exclusive reads (with at least one read traversing both variant positions) or in *cis* if variants occurred on the same reads (with at least one read traversing both variant positions).

### Identification of *RNU2-2* variants in 100kGP cases not included in AggV2

A list of genomes not included in AggV2 was generated using the 'rare_diseases_participant_disease', 'aggV2_gvcf_sample_stats' and 'genomes_and_file_paths' tables in LabKey. For samples aligned to GRCh38, all variants in *RNU2-2* with FILTER = = 'PASS' were retained. Samples were then filtered to those carrying either a homozygous variant or more than one heterozygous variant. For samples carrying more than one heterozygous variant, and for which at least one variant occurred in n.1–n.67 or the Sm site, phase was determined by manual inspection of reads in IGV.

For samples aligned to GRCh37, all variants in *RNU2-2* with FILTER = = 'PASS' were extracted. For samples carrying more than one heterozygous variant, all reads aligning to *RNU2-2* (plus 100 bp upstream and downstream) were extracted from BAM files. These reads were converted to FASTQ format using BEDtools (v.2.31.0)[53] and realigned using Bowtie2 (v.2.5.2)[54] to the GRCh38 FASTA sequence of *RNU2-2*. Only reads with a mapping quality of >30 were retained. The variant phase was determined by manual inspection of reads in IGV.

### Curation of biallelic variants in *RNU2-2* in control databases

Homozygous variants in gnomADv4 and the All of Us dataset were reviewed in respective online browsers (https://gnomad.broadinstitute.org and https://databrowser.researchallofus.org, respectively). Homozygous variants in *RNU2-2* were also identified in short-read genome sequencing data from 490,541 individuals in UKB[29]. Compound heterozygous variants were identified from a subset of 200,011 of these individuals for whom statistically phased genotype data were available[55].

### Variant position enrichment analysis

Biallelic genotypes were identified among our unsolved NDD cohort, 100kGP controls and controls in UKB as described above. The statistical analysis of variant distributions was limited to genotypes in which both alleles were entirely contained within the start and end coordinates of the canonical *RNU2-2* transcript (NR_199791.1) and in which both variants were either single nucleotide variants or indels less than 3 nt in length. The OR that genotypes included a variant within the 5′ constrained region (n.1–n.67) or the Sm site (n.97–n.107) of *RNU2-2* was calculated for cases versus controls. *P* values were determined from two-sided Fisher's exact tests.

### Cohort expansion

The GMS data were accessed through the genome_file_paths_and_types table in LabKey. Variants from 29,782 samples were extracted and filtered as described above for the 100kGP cohort. Compound heterozygous genotypes were identified as described above.

Solve-RD data ($n$ = 334 genomes) were accessed through the RD-CONNECT portal (https://rd-connect.eu). Variants were filtered using the Genotype–Phenotype Analysis Platform with a gnomAD minor allele frequency of <0.001 and for non_coding_exon_variant consequence. Individuals with two or more heterozygous genotypes were retained if at least one variant fell within n.1–n.67 or the Sm site of *RNU2-2*. Compound heterozygosity was inferred by manual inspection of reads in IGV.

The UDN-Aus dataset ($n$ = 249 genomes from 94 families) was accessed through the Centre for Population Genomics' rare disease genomic analysis platform, CaRDinal. Variants were filtered for those with a gnomADv4 minor allele frequency of <0.001 using seqr[32]. Variants were phased using parental data, and compound heterozygous or homozygous genotypes were retained if at least one variant fell within positions n.1–n.67 or the Sm site of *RNU2-2*. The South Korean, Swedish and Saudi Arabian datasets were filtered using these criteria also.

**Phenotype similarity analysis and enrichment of HPO terms**

HPO terms were extracted for all probands in our discovery cohort. Siblings ($n = 7$) were excluded to limit this analysis to only unrelated individuals ($n = 31$). Phenotype similarity was computed using the OntologySimilarity package[56]. We sampled 1,000 permutations of the same number ($n = 31$) of randomly selected unrelated individuals with NDD in 100kGP. Kernel density estimate and boxplots were plotted using ggplot2 in R, and the Monte Carlo $P$ value was calculated for the observed similarity statistic.

To calculate enrichment of HPO terms in the recessive condition, HPO terms were extracted for the 31 probands in our discovery cohort. The OntologyX[57] R package was used to determine HPO term frequencies in the cohort and remove redundant terms. This process was repeated for all unrelated individuals with NDD in 100kGP ($n = 10,157$). ORs for non-redundant terms between the two cohorts were calculated. Significance was determined with two-sided Fisher's exact tests followed by FDR correction.

**Clinical characterization of recessive *RNU2-2* disorder**

This research was performed under the ethical approvals given by the South Manchester NHS Research Ethics Committee (11/H1003/3/AM02). Written informed consent for the inclusion of detailed clinical information, imaging data and facial photographs was obtained from all participants or their parents.

Contact was made with recruiting clinicians through the Genomics England Airlock (Contact Clinicians Request). A standardized clinical proforma was circulated to all clinicians for completion.

**RNA sequencing and aberrant splicing analyses**

Whole blood samples were collected from a subset of 100kGP participants at the time of recruitment and stored in PaxGene tubes for future RNA sequencing. The protocol for RNA sequencing and data processing is publicly available at https://re-docs.genomicsengland.co.uk/rna_seq. In brief, RNA was extracted from whole blood samples from 5,546 probands with rare conditions. Samples were depleted of rRNA, and sequencing libraries were constructed using the Illumina Stranded Total RNA Prep with Ribo-Zero Plus protocol. Sequencing was performed with 2 × 100 bp paired-end reads on a NovaSeq 6000. Read mapping was performed using the Illumina DRAGEN RNA Pipeline v.3.8.4.

Splicing outlier analysis with FRASER2 (ref. 57) and gene expression outlier analysis with OUTRIDER[40] were performed for 5,412 samples in batches of 500 samples as part of the DROP pipeline[58]. Significant outlier events in FRASER2 were defined as those with an FDR-adjusted $P$ value of <0.1. Nominally significant outlier events were defined as those with an unadjusted $P$ value of <0.05. We identified five individuals with pathogenic heterozygous variants in *RNU2-2* and nine individuals with candidate biallelic variants in *RNU2-2* within this cohort. FRASER2 splicing outliers within these case sets were compared against 301 controls, defined as participants under 18 years of age with non-NDD phenotypes. Specifically, we defined individuals with non-NDD phenotypes as those not recruited under the 'Neurology and neurodevelopmental disorders' disease group in 100kGP (as per the 'rare_disease_participant_disease' table in LabKey) and who did not have any of the following HPO terms: HP:0000729 (autistic behavior), HP:0001250 (seizure), HP:0000252 (microcephaly), HP:0000750 (delayed speech and language development) and HP:0001263 (global developmental delay). Subsequently, these analyses were repeated for a subset of splicing outliers within known monogenic epilepsy genes with an 'amber' or 'green' review status on the genetic epilepsy panel in PanelApp Australia v.1.178.

To identify shared splicing events between *RNU2-2* variant carriers, the number of nominally significant FRASER2 outlier events observed in more than one individual was counted. The number of shared splicing events was then counted for 1,000 permutations of

an equal number of randomly selected controls ($n = 5$ vs heterozygous cases; $n = 9$ vs biallelic cases). This procedure was performed for each class of aberrant splicing event (annotated splice acceptor and splice donor, novel acceptor and annotated donor, novel donor and annotated acceptor, novel acceptor and novel donor). Significance was tested with two-sided percentile bootstrap $P$ values followed by Bonferroni correction for eight tests at $\alpha = 0.05$.

**Quantifying *RNU2-2* and *RNU2-1* expression**

Expression outlier analysis was initially conducted with OUTRIDER from the DROP pipeline run on 5,412 samples, as described above. Significant outlier events in OUTRIDER were defined as those with an FDR-adjusted $P$ value of <0.05.

To directly quantify *RNU2-2* and *RNU2-1* expression, reads aligning uniquely to U2-2 (located on chromosome 11) and U2-1 (including all 13 copies of *RNU2-1* located on chromosome 17 in the GRCh38 reference genome) were counted using Samtools[52]. RPM values for *RNU2-2*, *WDR74* and *RNU2-1* were obtained by normalizing read counts to the total number of unique reads aligning to chr11 or chr17, respectively. For *RNU2-2* and *RNU2-1*, this analysis was performed on 5,457 samples for which RNA sequencing data were available in BAM format. An identical procedure was performed for *WDR74* at a later time point on a slightly larger cohort of 5,544 samples. Two-tailed Mann–Whitney $U$-tests were used to compare RPM values and U2-2:U2-1 ratios between cases and controls.

**Comparison of phenotypes of dominant and recessive U2-related disorders**

Detailed clinical information, with written consent in place, was obtained through personal correspondence with the responsible clinician for 28 individuals with candidate biallelic variants in *RNU2-2*. For the dominant disorder, detailed clinical information was obtained from the supplementary materials of previous publications[10,11].

Phenotype terms were harmonized across all returned proformas. For principal components analysis, phenotypes were encoded as Boolean values as per prior work[15,59]. No distinction was made between different severities of the same phenotype (for example, global developmental delay). Missing values were coded as absent (that is, '0'). Principal component analysis was performed in R using the FactoMineR (v.2.13)[60] and factoextra (v.2.0.0)[61] packages.

The enrichment of phenotype terms between the recessive and dominant disorders was quantified with ORs. ORs were only calculated for phenotypes observed in at least five individuals in either the dominant or recessive cohorts. Where phenotype counts included zero values, the Haldane–Anscombe correction was applied. Statistical significance was determined with two-sided Fisher's exact tests with FDR correction.

**Estimating the frequency of the recessive *RNU2-2* disorder and its comparison with other NDDs**

Genomic Medicine Centre exit questionnaires were accessed from the gmc_exit_questionnaire table in LabKey within the GERE. NDD cases were defined using the rare_disease_participant_phenotype table in LabKey, retaining all individuals who had phenotype terms matching those used for the enrichment analysis described above. The exit questionnaire data were filtered for these individuals only. The exit questionnaire data were also filtered for genes with 'green' review status within the intellectual disability panel (R29) in PanelApp[42]. The number of families with a confirmed diagnosis attributed to each gene was obtained. The number of *RNU4-2*-related ReNU syndrome diagnoses was determined by counting variants in the 18 bp critical region reported in a previous publication[43] for all individuals with NDD. The number of dominant *RNU2-2* diagnoses was determined by counting the recurrent n.4 G > A, n.35 A > G and n.35 A > C variants among all individuals with NDD. The number of *RNU4-2* recessive diagnoses was taken from the prioritized 100kGP biallelic variant carriers described in a previous publication[15].

## Reporting summary

Further information on research design is available in the Nature Portfolio Reporting Summary linked to this article.

## Data availability

Genomic and phenotypic data are available for the 100kGP and individuals who have had whole genome sequencing through the GMS in the NGRL. Access to the NGRL may be granted following application at https://www.genomicsengland.co.uk/research/academic/join-research-network, which gives access to the secure GERE. Genomic data used pertain to participants in 100kGP in the Main Programme v.19 and the GMS data v.4. RNA sequencing data derived from blood are accessible through GERE, with details found in LabKey. Solve-RD data are accessible by application through the RD-CONNECT platform. All data presented in this paper pertaining to 100kGP participants were requested for the Airlock transfer through GERE. The paper was submitted for approval by the Genomics England Publication Committee on 25 August 2025 and was approved on 27 August 2025. Access to the Australian Centre for Population Genomics dataset can be requested through contact with the authors. The GRCh38 human genome reference assembly can be accessed at https://www.ncbi.nlm.nih.gov/datasets/genome/GCF_000001405.26. GENCODE v.32 comprehensive annotations were accessed within the GERE but can be downloaded from https://www.gencodegenes.org/human/release_32.html. The gnomADv4 genotype VCF files were accessed within the GERE but can also be downloaded from https://gnomad.broadinstitute.org. Source data are provided with this paper.

## Code availability

Software packages bedtools (v.2.31.0), bcftools (v.1.16) and samtools (v.1.9) were the predominant tools used in this study. R (v.4.1.1) was used in RStudio with plots generated using ggplot (v.3.5.2) and related packages. Code generated for analyses is stored securely in the GERE and available to be shared within this environment upon request.

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

## Acknowledgements

We thank the participants and the recruiting clinicians of the 100kGP. We also thank Genomics England for generating the data and providing the GERE platform for analysis. Specific thanks to P. O'Donovan, M. Sato and Z. Mustafa for airlock requests; M. Hoti, A. L. Taylor and J. Yang for facilitating clinical collaboration requests; and M. McEntagart, C. Smith and N. Elkhateeb at Genomics England for additional support. We thank J. De Jonghe for his suggestions on the secondary structure illustration. Variant identification in the Swedish LoqusDB cohort was performed by staff from Clinical Genomics Stockholm, Science for Life Laboratory and Karolinska University Laboratory, which maintains the LoqusDB variant frequency database for national rare disease diagnostics. A.J. and S.B. acknowledge the support of Solve-RD. The Solve-RD project has received funding from the European Union's Horizon 2020 research and innovation program under grant agreement 779257. This study has been delivered through the National Institute for Health and Care Research (NIHR) Manchester Biomedical Research Centre (NIHR203308). S.B. acknowledges the support of the MRC Epigenomics of Rare Diseases (EpiGenRare) Node (MR/Y008170/1). A.J.M.B. is supported by a Wellcome PhD Training Fellowship for Clinicians and the 4Ward North PhD Programme for Health Professionals (223521/Z/21/Z). N.W. is supported by a Wellcome Career Development Award (grant no. 305292/Z/23/Z) and a Lister Institute research prize. We gratefully acknowledge the participants of the National Genomic Research Library (NGRL), whose contributions made this research possible. Secure access to the NGRL under project ID 961 was provided by Genomics England, which delivers the NGRL in partnership with NHS England and is wholly owned by the UK Department of Health and Social Care. The NGRL contains participants' health data collected by the NHS as part of their care, along with samples and data from their participation in research, for which fully informed consent has been obtained. This includes genomic and clinical data provided through the NHS Genomic Medicine Service, as well as data obtained through research studies, including the 100,000 Genomes Project and the Generation Study, both of which are delivered in partnership with the NHS, and from other research cohorts involving external collaborators. The Wellcome Trust, Cancer Research UK and the Medical Research Council have also funded research infrastructure. This research has been conducted using the UK Biobank Resource under application number 90952. S.M.S. is supported by the Epilepsy Society. The Australian Undiagnosed Diseases Network (UDN-Aus) acknowledges financial support from the Australian Government's Medical Research Future Fund (2007567), Australian Genomics and The Centre for Population Genomics (Garvan Institute of Medical Research and Murdoch Children's Research Institute), funded in part by a Medical Research Future Fund (MRFF) Genomics Health Futures Mission grant (2008820). E.E.P. is supported by a National Health and Medical Research Council (NHMRC) Investigator Grant (2021/GNT2008166). The Swedish Undiagnosed Diseases Network (UDN Sweden) acknowledges financial support from the Swedish Ministry of Health and Social Affairs. A.N. received support from Region Stockholm (5010124 ALF, 520136 ALF) and the Swedish Research Council (2021-02860). A.W. acknowledges the Knut and Alice Wallenberg Foundation, KAW2020.0228. Many authors of this publication are members of the European Reference Network on Rare Congenital Malformations and Rare Intellectual Disability ERN-ITHACA. ERN-ITHACA is funded by the European Union, under grant agreement no. 101156387.

## Author contributions

A.J., A.J.M.B. and S.B. conceptualized the study, analyzed data and wrote the manuscript. S.B. and J.L. provided supervision. B.A., K.B.,

J.E., O.J.H., N.P.P., N.L., T.S., A.M.D.-V., A.W., A.L., A.H., F. Taylan, A.N., O.A., N.D., J.L., R.R., C.S. and S. Walker analyzed data. A.J., S.B., A.N., A.M.D.-V., K.K., T.S., N.P.P., S.A., E.B., A.F.B., H.B., K.E.C., N.C., F.E., L.J.E., A.P.F., G.G., U.K., R.M., S.M., L.M., L.N., S.P.H., K.M., A.M., E.E.P., A.P., K.P., M.R., A.R., C.L.R., R.S., S.M.S., H.S., S. Wiafe, F. Tinella, E.W., M.D., I.D., J.H.C. and F.A. assisted in clinical data collection. C.F., B.B.-P., R.T.O'K. and H.B.T. generated data. N.W. provided technical expertise. All authors read and approved the final version of the manuscript before submission.

## Competing interests

L.M. has received personal consultancy fees from Mendelian, a rare disease digital health company, outside of the submitted work. N.W. has received research grant funding from Novo Nordisk and BioMarin Pharmaceutical. All other authors declare no competing interests.

## Additional information

**Extended data** is available for this paper at https://doi.org/10.1038/s41588-026-02551-9.

**Correspondence and requests for materials** should be addressed to Adam Jackson or Siddharth Banka.

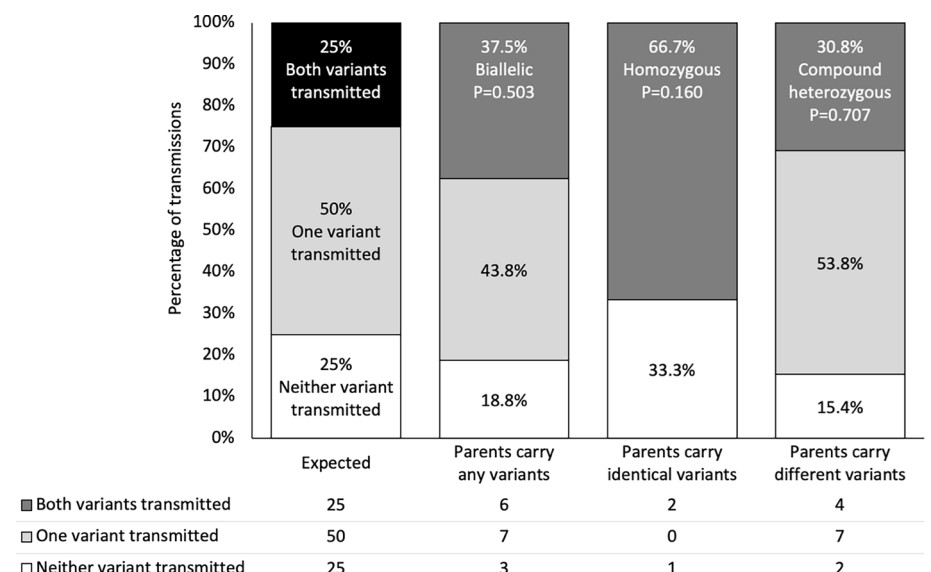

| | Expected | Parents carry any variants | Parents carry identical variants | Parents carry different variants |
|---|---|---|---|---|
| ■ Both variants transmitted | 25 | 6 | 2 | 4 |
| □ One variant transmitted | 50 | 7 | 0 | 7 |
| □ Neither variant transmitted | 25 | 3 | 1 | 2 |

**Extended Data Fig. 1 | Transmission analysis of biallelic *RNU2-2* variants in 100KGP.** Stacked bar plot showing transmission of heterozygous *RNU2-2* variants from parents to offspring among 100kGP controls.

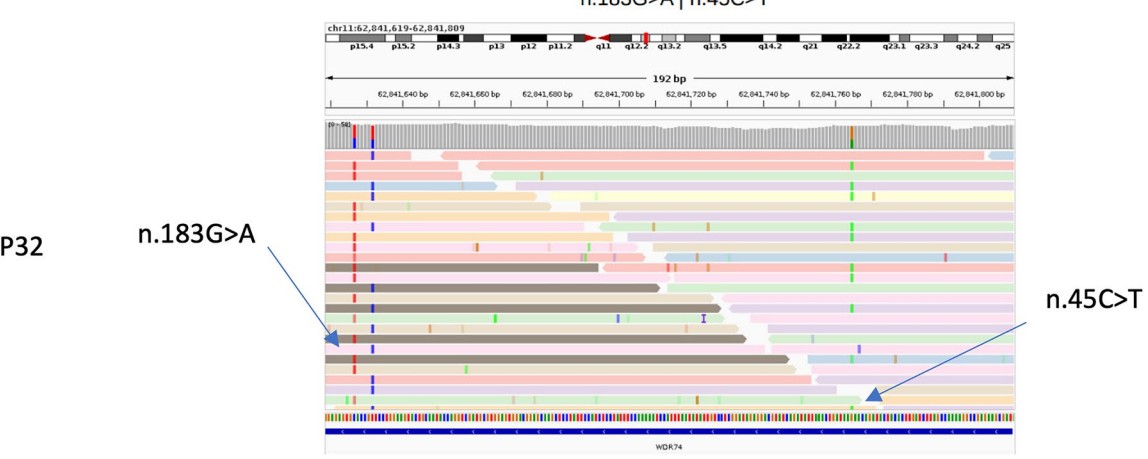

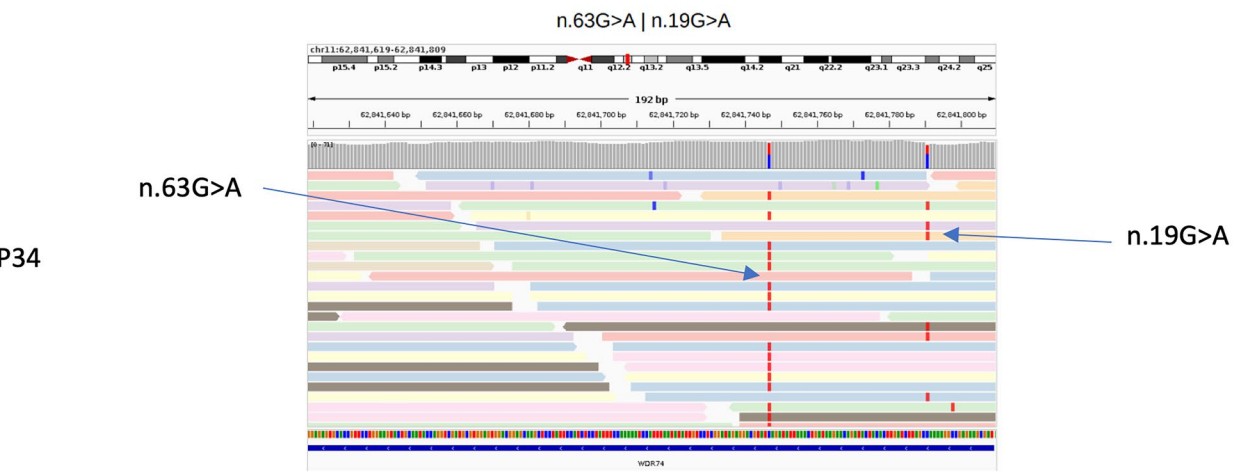

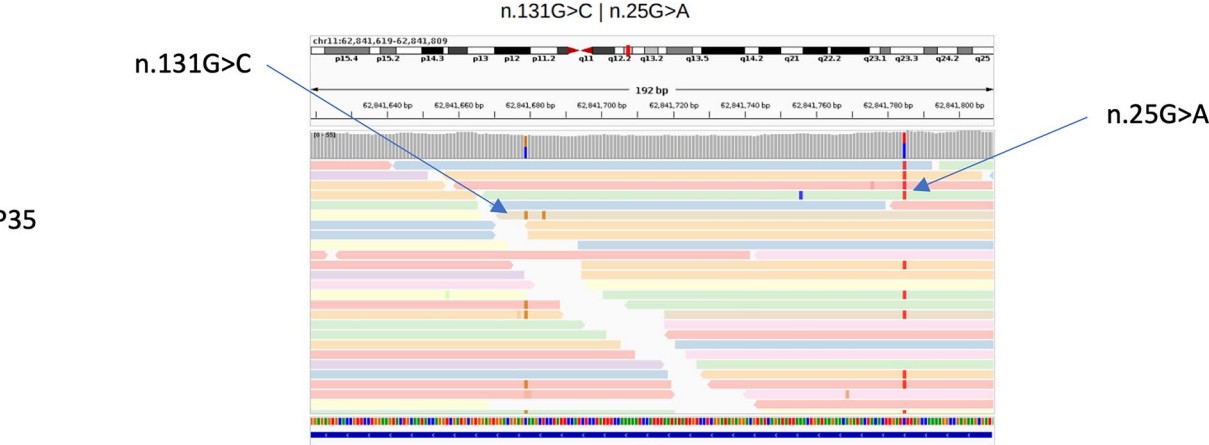

**Extended Data Fig. 2 | Identifying phase switch errors in the aggregated 100KGP dataset.** IGV screenshots of sequencing reads covering *RNU2-2* for individuals with unsolved NDD. Screenshots for three individuals with "phase switch" errors (false negative for compound heterozygosity) in aggV2 are shown. In the top panel, due to the large distance between variants, there is only one informative read (the lowest read in the IGV plot).

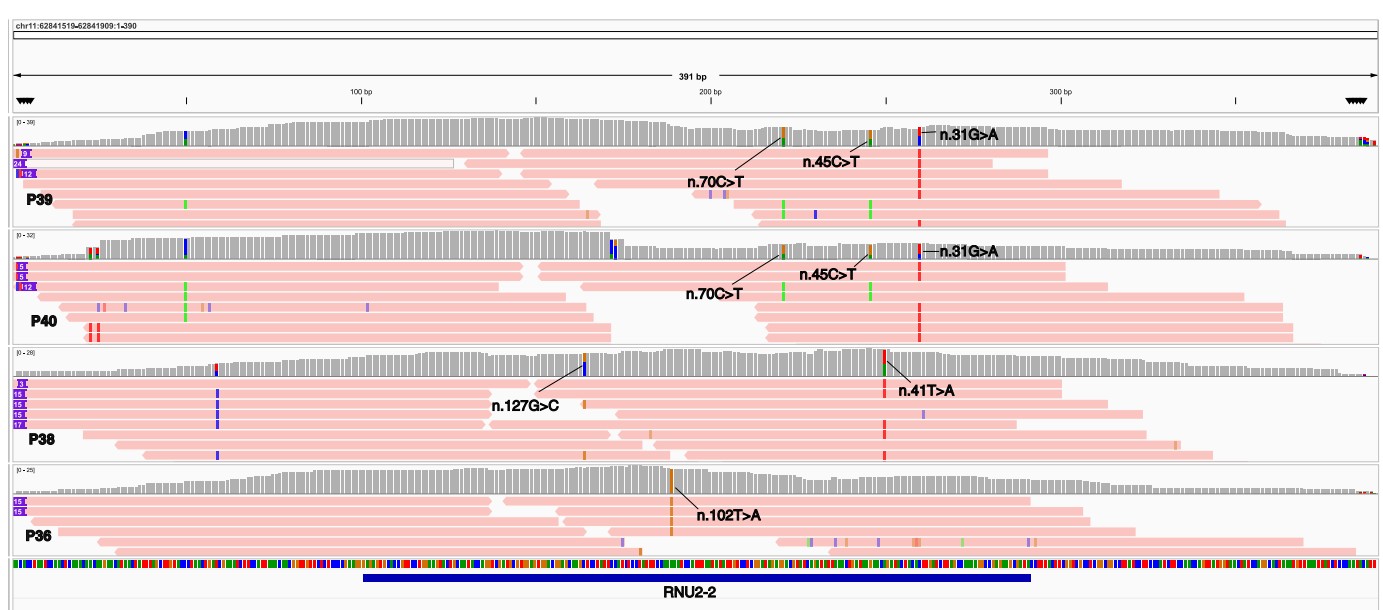

**Extended Data Fig. 3 | Re-mapping reads aligned to GRCh37 to GRCh38 in order to confidently variant call in *RNU2-2*.** IGV screenshots of sequencing reads covering *RNU2-2* for individuals with unsolved NDD who had their data aligned to GRCh37, where *RNU2-1* locus is not mapped. The IGV screenshot shows all reads re-mapped to *RNU2-2* in GRCh38, indicating confident variant calls. For P39 and P40, the rare (n.45 C > T and n.31 G > A) variants are shown, while a common variant (n.70 C > T) is in linkage with n.45 C > T.

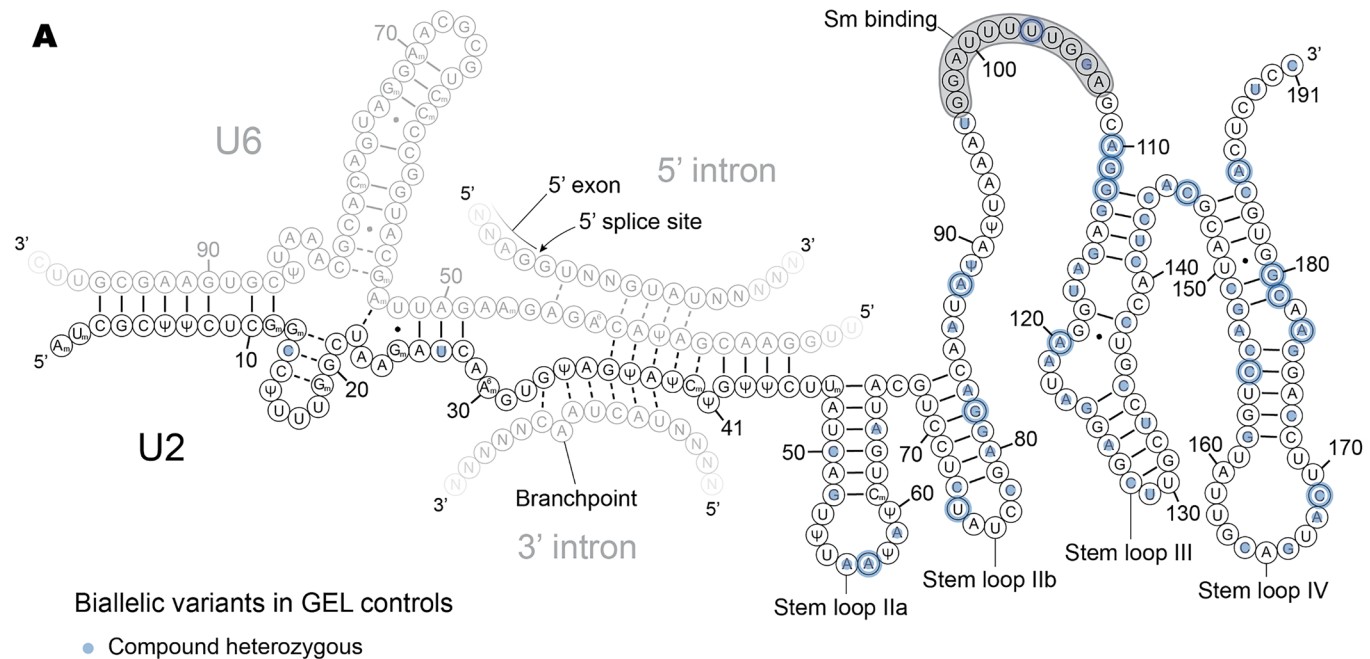

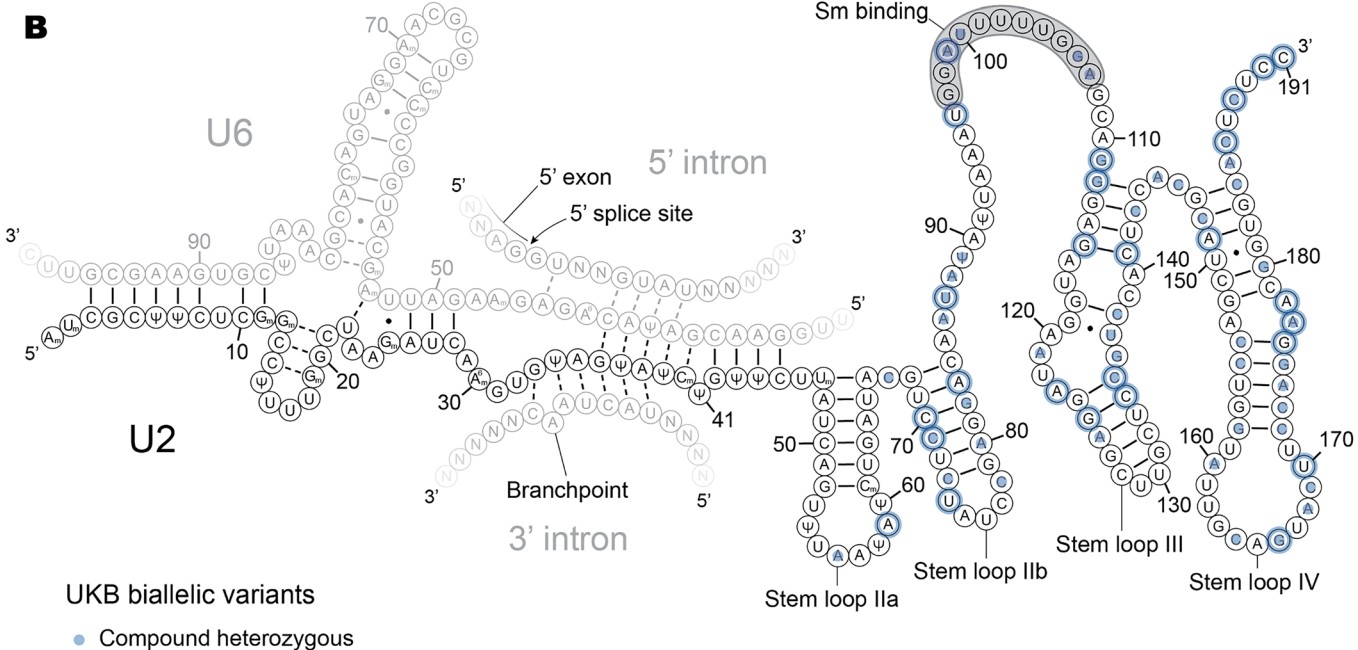

**Extended Data Fig. 4 | Secondary structure of U2-2 in complex with U6, with biallelic variants in control cohorts overlaid. a**, Biallelic variants in GEL controls. **b**, Biallelic variants in UK Biobank. Secondary structure annotations are shown as in Fig. 2b.

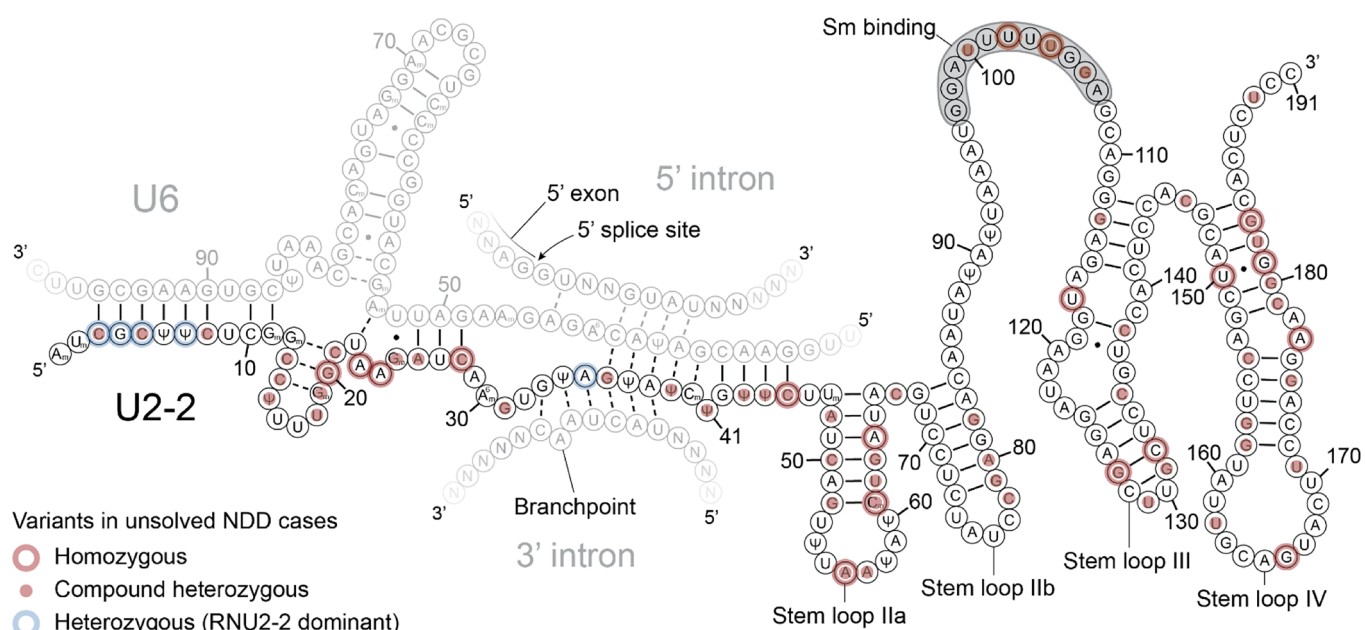

**Extended Data Fig. 5 | Secondary structure of U2-2 in complex with U6, the 5' splice site, and the splicing branchpoint.** The proposed four-helix U2/U6 structure described by Karunatilaka and Rueda is shown. Red markers show candidate biallelic variants in unsolved NDD cases. Blue markers show heterozygous variants pathogenic for the dominant *RNU2-2* disorder. The Sm binding site is highlighted in grey. Solid lines show stable Watson-Crick base pairs. Dashed lines show transient Watson-Crick base pairs. Black points show wobble base pairs. 2'-O-methylated nucleotides are depicted by $N_m$, pseudouridine by Ψ, and N$^6$-methylated adenosine by $A^6_m$. BPRS, branchpoint recognition sequence.

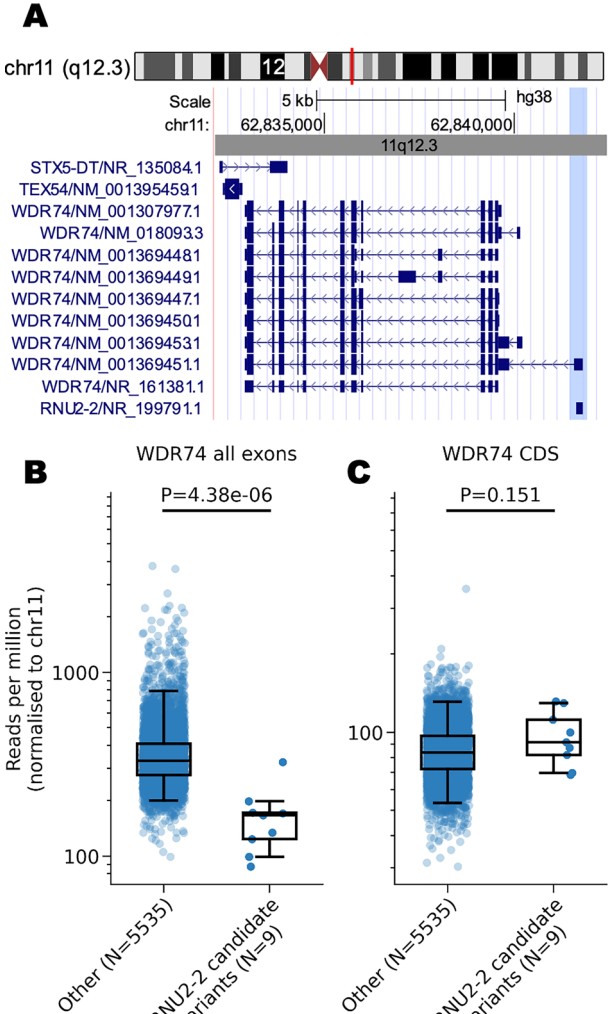

**Extended Data Fig. 6 | WDR74 expression in carriers of candidate biallelic variants in RNU2-2. a**, Screengrab from the UCSC genome browser showing the *WDR74* locus and adjacent genomic context (GRCh38 chr11:62,832,122-62,842,440). The blue highlight shows the overlap of *RNU2-2* with the 5′ UTR of a single transcript in *WDR74* (NM_001369451.1). **b**, Relative expression of *WDR74* in individuals with candidate biallelic variants in *RNU2-2* versus controls. Reads mapping to all exons in *WDR74* are shown. **c**, As in **b**, but for reads mapping to CDS exons in *WDR74*. For **b** and **c**, read counts are normalized to chromosome 11. *P* values from two-tailed Mann-Whitney U tests are shown. Box and whisker plots show the median, quartiles, and +/- 1.5 times the interquartile range of the data.

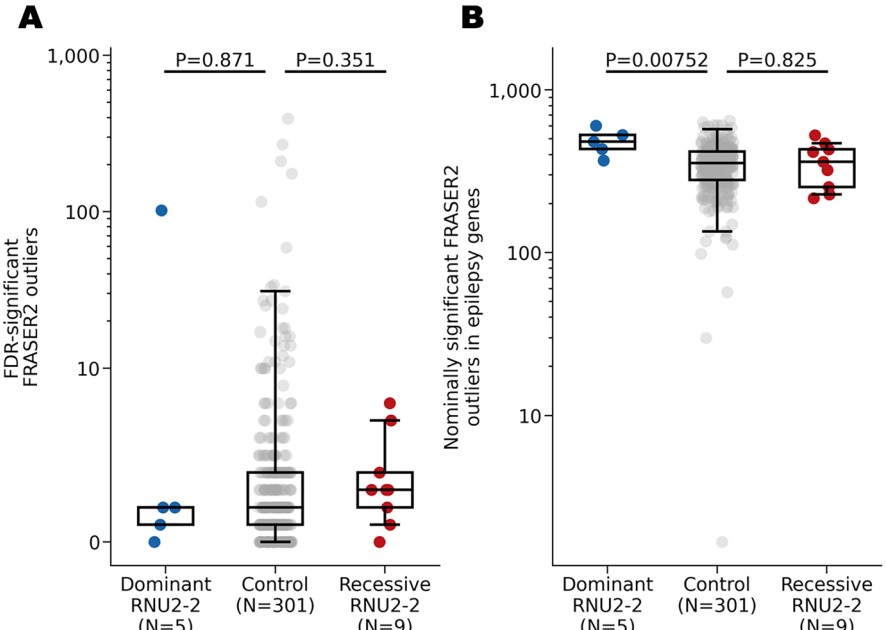

**Extended Data Fig. 7 | Aberrant splicing events from RNA sequencing data in individuals with candidate variants in *RNU2-2* versus controls. a**, FDR-significant *(FDR P < 0.10)* FRASER2 splicing outliers. **b**, Nominally significant (unadjusted *P < 0.05*) FRASER2 outliers in known monogenic epilepsy genes from PanelApp Australia. In each panel, data are shown for individuals with heterozygous variants pathogenic for the dominant *RNU2-2* disorder (blue circles) (*n* = 5), non-NDD controls (grey circles) (*n* = 301), and individuals with unsolved NDD and candidate biallelic variants in *RNU2-2* (red circles) (*n* = 9). *P* values from two-tailed Mann-Whitney U tests are shown. Box and whisker plots show the median, quartiles, and +/- 1.5 times the interquartile range of the data.

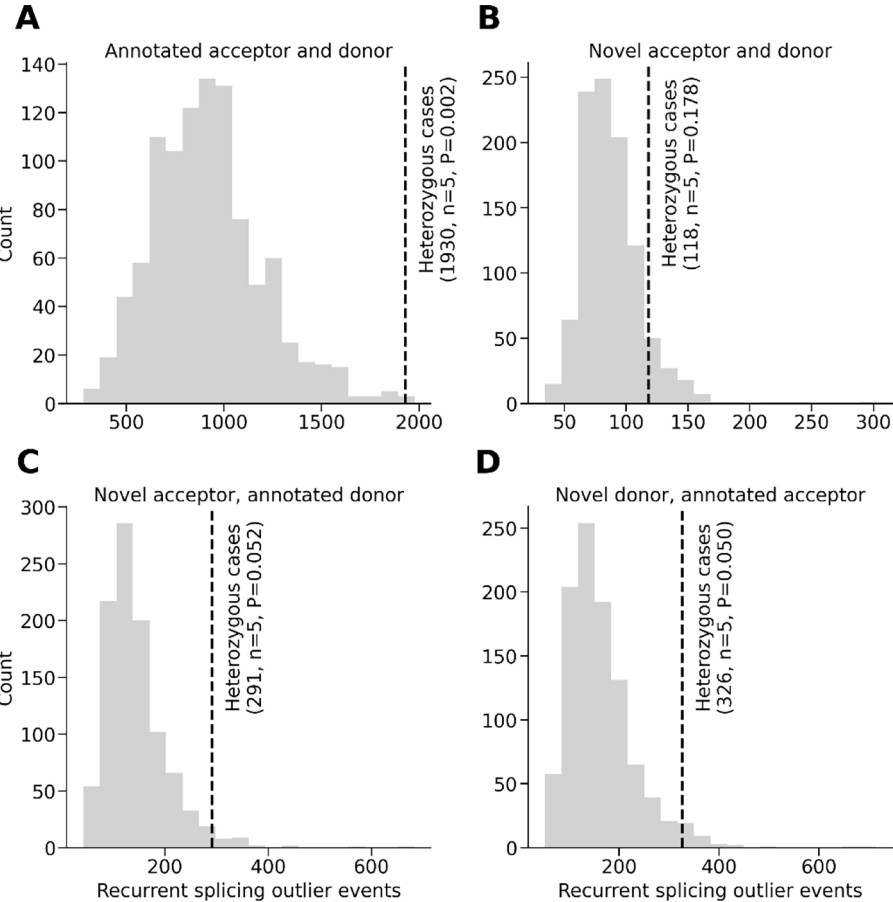

**Extended Data Fig. 8 | Recurrent splicing events in individuals with heterozygous variants pathogenic for the dominant *RNU2-2* disorder.**
**a**, Aberrant splicing events at annotated splice junctions. **b**, Novel splice donor and acceptor sites. **c**, Splice junctions containing novel splice acceptor sites and annotated splice donor sites. **d**, Novel splice donor sites and annotated splice acceptor sites. Histograms show the number of identical nominally significant FRASER2 aberrant splicing events observed in >1 individual from 1,000 random samples of 5 controls. The dashed black lines show the number of identical nominally significant FRASER2 aberrant splicing events observed in >1 individual among 5 cases with pathogenic heterozygous variants in *RNU2-2*. The number of recurrent splicing events is given in parentheses. Two-sided bootstrap *P* values are shown with correction for multiple testing.

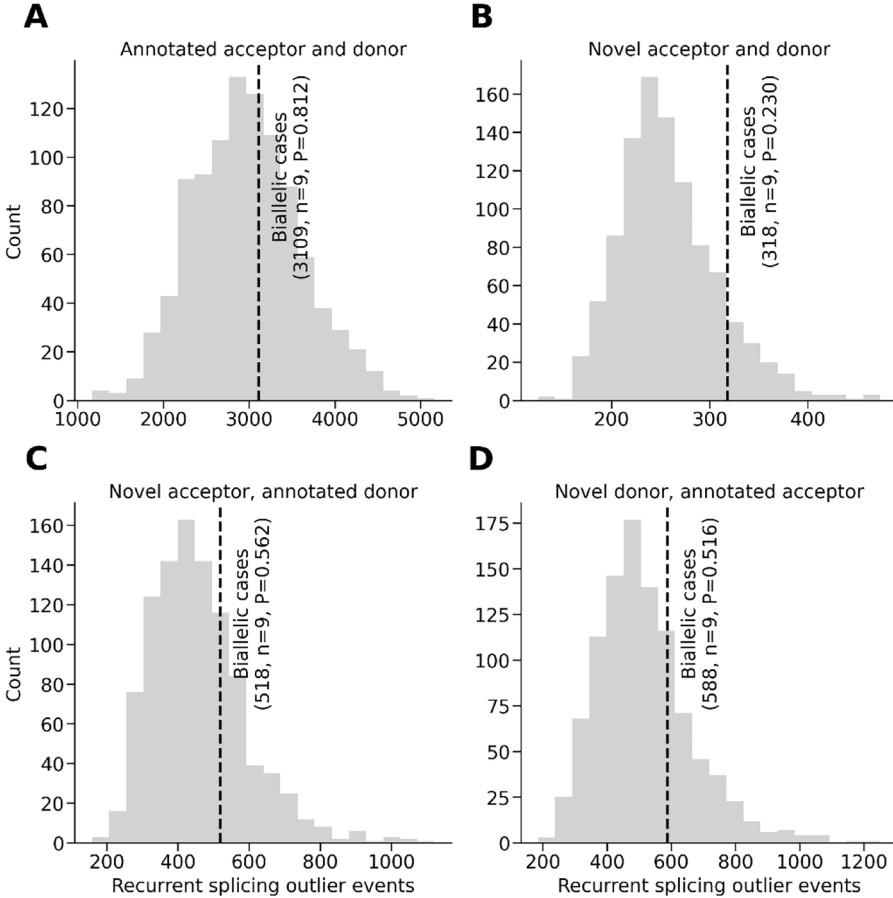

**Extended Data Fig. 9 | Recurrent splicing events in individuals with candidate biallelic variants in *RNU2-2*. a-d**, Plots arranged as in Extended Data Fig. 8. Histograms show the number of identical nominally significant FRASER2 aberrant splicing events observed in >1 individual from 1,000 random samples of 9 controls. The dashed black lines show the number of identical nominally significant FRASER2 aberrant splicing events observed in >1 individual among 9 cases with candidate biallelic variants in *RNU2-2*. The number of recurrent splicing events is given in parentheses. Two-sided bootstrap *P* values are shown without correction for multiple testing.

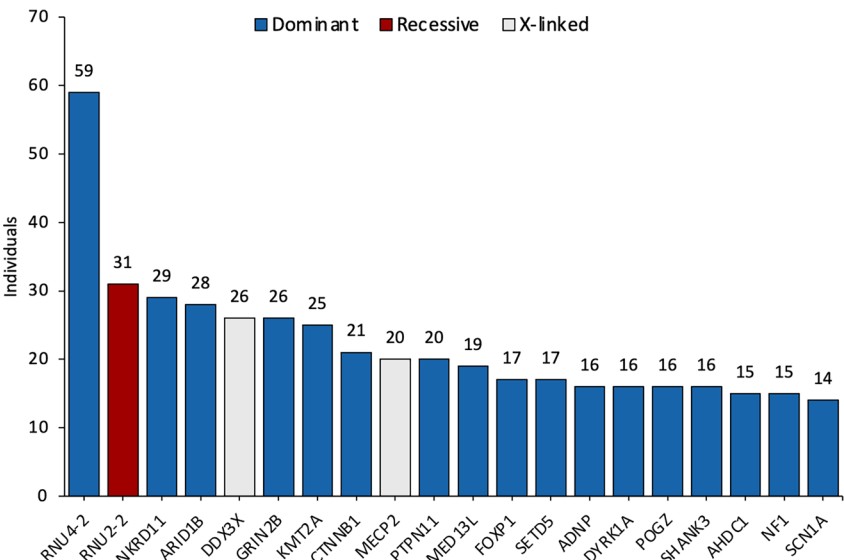

**Extended Data Fig. 10 | The contribution of variants in snRNAs to unsolved NDDs.** Bar plot showing the 20 most frequent diagnoses on the R29 Intellectual Disability panel for individuals in the 100kGP recruited with NDD (*n* = 10,987). Counts for the *RNU2-2* and *RNU4-2* disorders were based on genotyping (see Methods). Counts for all other genes were taken from exit questionnaire data. Genes causal for dominant disorders are shown in blue, for X-linked disorders in grey, and for recessive disorders in red.

# Reporting Summary

## Statistics

For all statistical analyses, confirm that the following items are present in the figure legend, table legend, main text, or Methods section.

| n/a | Confirmed | |
|---|---|---|
| ☐ | ☒ | The exact sample size (*n*) for each experimental group/condition, given as a discrete number and unit of measurement |
| ☐ | ☒ | A statement on whether measurements were taken from distinct samples or whether the same sample was measured repeatedly |
| ☐ | ☒ | The statistical test(s) used AND whether they are one- or two-sided<br>*Only common tests should be described solely by name; describe more complex techniques in the Methods section.* |
| ☐ | ☒ | A description of all covariates tested |
| ☐ | ☒ | A description of any assumptions or corrections, such as tests of normality and adjustment for multiple comparisons |
| ☐ | ☒ | A full description of the statistical parameters including central tendency (e.g. means) or other basic estimates (e.g. regression coefficient) AND variation (e.g. standard deviation) or associated estimates of uncertainty (e.g. confidence intervals) |
| ☐ | ☒ | For null hypothesis testing, the test statistic (e.g. *F*, *t*, *r*) with confidence intervals, effect sizes, degrees of freedom and *P* value noted<br>*Give P values as exact values whenever suitable.* |
| ☒ | ☐ | For Bayesian analysis, information on the choice of priors and Markov chain Monte Carlo settings |
| ☒ | ☐ | For hierarchical and complex designs, identification of the appropriate level for tests and full reporting of outcomes |
| ☐ | ☒ | Estimates of effect sizes (e.g. Cohen's *d*, Pearson's *r*), indicating how they were calculated |

*Our web collection on statistics for biologists contains articles on many of the points above.*

## Software and code

Policy information about availability of computer code

| Data collection | Data was accessed in the National Genomics Research Library (NGRL) for all individuals in the 100,000 Genomes Project. This included variant data from whole genome sequencing and also Human Phenotype Ontology terms. Recruitment categories and also family relationships and sequencing structures were also derived from this resource. The NGRL also houses the GMS data which was used as a validation cohort in this manuscript.<br><br>Other validation cohorts included Solve-RD, UDNAus, South Korean Undiagnosed Diseases Database, Saudi Arabia Lifera Database and Swedish Undiagnosed Diseases Network, which were accessed through either the RD-CONNECT platform of personal communication with co-authors. |
|---|---|
| Data analysis | Software used for data analysis are housed within the NGRL and include samtools (v1.10), bftools (v1.16), bowtie2 (v2.5.2), bedtools (v.2.31.0), FRASER2, OUTRIDER. |

For manuscripts utilizing custom algorithms or software that are central to the research but not yet described in published literature, software must be made available to editors and reviewers. We strongly encourage code deposition in a community repository (e.g. GitHub). See the Nature Portfolio guidelines for submitting code & software for further information.

## Data

Policy information about <u>availability of data</u>

All manuscripts must include a <u>data availability statement</u>. This statement should provide the following information, where applicable:

- Accession codes, unique identifiers, or web links for publicly available datasets
- A description of any restrictions on data availability
- For clinical datasets or third party data, please ensure that the statement adheres to our <u>policy</u>

Genomic and phenotypic data are available for the 100KGP and individuals who have had WGS through the Genomic Medicine Service in the NGRL. Access to the NGRL may be granted following application via https://www.genomicsengland.co.uk/research/academic/join-research-network, which gives access to the secure GERE. Genomic data used pertain to participants in 100KGP in the Main Programme v.18 and the GMS data v.4. SolveRD data are accessible by application through the RD-CONNECT platform. All data presented in this paper, pertaining to 100kGP participants, were requested for the Airlock transfer through GERE. The paper was submitted for approval by the Genomics England Publication Committee on 25th August 2025 and was approved on 27th August 2025. Access to the Australian Centre for Population Genomics dataset can be requested through contact with the authors. The GRCh38 human genome reference assembly can be accessed at https://www.ncbi.nlm.nih.gov/datasets/genome/GCF_000001405.26/. The GENCODE v.32 comprehensive annotations were accessed within the GERE but can be downloaded from https://www.gencodegenes.org/human/release_32.html. The gnomADv4 genotype VCF files were accessed within the GERE but can also be downloaded from https://gnomad.broadinstitute.org/.

## Research involving human participants, their data, or biological material

Policy information about studies with <u>human participants or human data</u>. See also policy information about <u>sex, gender (identity/presentation), and sexual orientation</u> and <u>race, ethnicity and racism</u>.

| | |
|---|---|
| Reporting on sex and gender | Sex is reported for all individuals with prioritized variants, which is founded in their recruitment sex. The disorder occurs with equal prevalence in both sexes and hence sex-specific analyses were not undertaken. |
| Reporting on race, ethnicity, or other socially relevant groupings | Where available, ethnicity is reported for individuals with detailed phenotype data who have provided additional consent. This refers to their self described ancestry / country / region of origin, rather than race. |
| Population characteristics | The main analysis of this study was undertaken in NGRL which houses genome sequencing data for over 100 000 individuals who have been recruited as either probands or relatives of probands with a rare disease. These are predominantly children and young adults with respect to neurodevelopmental disorders but we have not stipulated any age cut off in our analysis. Validation cohorts consist of individuals and their families with rare conditions. Not all cohorts stipulated neurodevelopmental disorders as a recruitment criteria although this is a common recruitment category in all. |
| Recruitment | Participants were prospectively recruited to the 100KGP and the same is the case for the GMS data. Recruitment was led by the Genomic Medicine Services in the UK. Recruitment was prospective for all validation cohorts also. |
| Ethics oversight | This research was performed under the ethical approvals given by the South Manchester National Health Service (NHS) Research Ethics Committee (REC; 11/H1003/3/AM02). Written informed consent for the inclusion of detailed clinical information, imaging data, and facial photographs, was obtained from all participants or their parents. |

Note that full information on the approval of the study protocol must also be provided in the manuscript.

# Field-specific reporting

Please select the one below that is the best fit for your research. If you are not sure, read the appropriate sections before making your selection.

☒ Life sciences    ☐ Behavioural & social sciences    ☐ Ecological, evolutionary & environmental sciences

For a reference copy of the document with all sections, see <u>nature.com/documents/nr-reporting-summary-flat.pdf</u>

# Life sciences study design

All studies must disclose on these points even when the disclosure is negative.

| | |
|---|---|
| Sample size | Sample size was determined by the number of available genomes in the NGRL which is steadily increasing with specific data freezes. No power calculations were performed as the number of cases and controls in the NGRL is finite. |
| Data exclusions | For trio analyses looking at transmission of RNU2-2 variants, data aligned to GRCh37 was excluded as RNU2-1 locus is not annotated in this genome assembly giving rise to mapping errors which could affect the resultant data and analysis. |
| Replication | Replication was undertaken by consulting independent rare disease genome sequencing cohorts from around the world. Results were validated in all datasets analysed (GMS, SOLVE-RD, UDNAus, Sweden, Saudi Arabia and South Korea) |
| Randomization | Randomization was not appropriate as phenotypes needed to be grouped. |

| Blinding | For statistical analysis, authors were blind to which participants were in the neurodevelopmental disorders group and those who were in the control groups. For phenotype analysis, authors could not be blinded to participant ID and this would be inappropriate for such analyses. |
|---|---|

# Reporting for specific materials, systems and methods

We require information from authors about some types of materials, experimental systems and methods used in many studies. Here, indicate whether each material, system or method listed is relevant to your study. If you are not sure if a list item applies to your research, read the appropriate section before selecting a response.

## Materials & experimental systems

| n/a | Involved in the study |
|---|---|
| ☒ ☐ | Antibodies |
| ☒ ☐ | Eukaryotic cell lines |
| ☒ ☐ | Palaeontology and archaeology |
| ☒ ☐ | Animals and other organisms |
| ☒ ☐ | Clinical data |
| ☒ ☐ | Dual use research of concern |
| ☒ ☐ | Plants |

## Methods

| n/a | Involved in the study |
|---|---|
| ☒ ☐ | ChIP-seq |
| ☒ ☐ | Flow cytometry |
| ☒ ☐ | MRI-based neuroimaging |

## Plants

| Seed stocks | n/a |
|---|---|

| Novel plant genotypes | n/a |
|---|---|

| Authentication | n/a |
|---|---|

