## [Peer Review File · Nature Genetics]

Biallelic variants in RNU2-2 cause a remarkably frequent developmental and epileptic encephalopathy

Corresponding Author: Professor Siddharth Banka

Version 0:

Decision Letter:

14th October 2025

Dear Sid,

Your Article "Biallelic variants in RNU2-2 cause a remarkably frequent developmental epileptic encephalopathy" has been seen by three referees. You will see from their comments below that, while they find your work of interest, they have raised several relevant points. We are interested in the possibility of publishing your study in Nature Genetics, but we would like to consider your response to these points in the form of a revised manuscript before we make a final decision on publication.

To guide the scope of the revisions, the editors discuss the referee reports in detail within the team, including with the chief editor, with a view to identifying key priorities that should be addressed in revision and sometimes overruling referee requests that are deemed beyond the scope of the current study. In this case, we ask that you revise specific aspects of the presentation for clarity, providing additional details and adjusting interpretations where needed, and that you extend the discussion as requested by the reviewers, in particular placing your findings in context with the related preprints on medRxiv. We hope you will find this prioritized set of referee points to be useful when revising your study. Please do not hesitate to get in touch if you would like to discuss these issues further.

We therefore invite you to revise your manuscript taking into account all reviewer and editor comments. Please highlight all changes in the manuscript text file. At this stage, we will need you to upload a copy of the manuscript in MS Word .docx or similar editable format.

*2) If you have not done so already, please begin to revise your manuscript so that it conforms to our Article format instructions, available

[here](http://www.nature.com/ng/authors/article_types/index.html).

*3) Include a revised version of your Reporting Summary: <https://www.nature.com/documents/nr-reporting-summary.pdf> It will be available to referees (and, potentially, statisticians) to aid in their evaluation if the manuscript goes back for peer review.

Please be aware of our [guidelines](https://www.nature.com/nature-research/editorial-policies/image-integrity) on digital image standards.

EXTENDED DATA FIGURES

When re-submitting your manuscript, please ensure that any supplementary figures and tables that are crucial to the

manuscript's conclusions are converted into Extended Data figures and tables to increase visibility of these data. Extended Data figures and tables are online-only (present in the online PDF and full-text HTML versions of the paper), peer-reviewed display items that provide essential background to the article but are not included in the main article due to space constraints. A maximum of ten Extended Data display items (figures and tables) is permitted.

Link Redacted

We hope to receive your revised manuscript within 4-8 weeks. If you cannot send it within this time, please let us know.

Nature Genetics is committed to improving transparency in authorship. As part of our efforts in this direction, we are now requesting that all authors identified as 'corresponding author' on published papers create and link their Open Researcher and Contributor Identifier (ORCID) with their account on the Manuscript Tracking System (MTS), prior to acceptance. ORCID helps the scientific community achieve unambiguous attribution of all scholarly contributions. You can create and link your ORCID from the home page of the MTS by clicking on 'Modify my Springer Nature account'. For more information, please visit www.springernature.com/orcid.

Sincerely,
Kyle

Kyle Vogan, PhD
Senior Editor
Nature Genetics
<https://orcid.org/0000-0001-9565-9665>

Referee expertise:

Referee #1: Genetics, neurodevelopmental disorders

Referee #2: Genetics, neurodevelopmental disorders

Referee #3: Genetics, neurodevelopmental disorders

Reviewers' Comments:

Reviewer #1 (Remarks to the Author):

* Summary of the key results

A novel genomic cause of NDD: autosomal recessive form of variants in the RNU2-2 snRNA gene. The resulting phenotype has distinctive features from that of the dominant RNU2-2-related syndrome. There is reduced abundance of the U2-2 RNA in the recessive form. Apparently, the most frequent recessive cause of NDD in the UK population.

* Originality and significance: if not novel, please include reference

The conclusions of the study are novel; however, the authors should acknowledge that similar conclusions have been published in preprints from other groups:

medRxiv 2025.08.26.25334179; doi: <https://doi.org/10.1101/2025.08.26.25334179>

medRxiv 2025.09.02.25334923; doi: <https://doi.org/10.1101/2025.09.02.25334923>

If some conclusions are different from those of the other studies, please mention them in the discussion.

* Data & methodology: validity of approach, quality of data, quality of presentation

Please better clarify the cohorts of patients used, the number of individuals versus number of families with biallelic variants, and the filtering in the different steps, particularly in the discovery cohort. Please also provide more info on the phenotypes in the validation cohorts and the filtering that led to the final 14 additional families. The authors tried to clarify the discovery flow in figure 1C, but I confess its content is not clear. I suggest to rework this figure and its legend.

An important point: Please mention if the same patient cohorts have been used by the authors of the competing and

published (in preprints) studies.

L229-238 not very clear. What was the rationale in L232-233? (For compound heterozygous variants, we only included those genotypes in which at least one variant fell in the 5' constrained region (n.1-n.67) or the Sm site (n.97-n.107)). Is it possible that some true positives were filtered out?

L246 31? there are 38 in L226

L293 delete RNU2-2

L292 Why 6 individuals without RNU2-2 biallelic variants were outliers for U2-2 RNA levels? Hypotheses?

L323-324 The argument that the ratio U2-2/U2-1 is a better marker than the U2-2 RNA abundance is not convincing. Figures 4A and 4C do not seem to support this.

Why the RNA data were not normalized to the sum of reads for chromosomes 11 and 17 (or to the whole transcriptome)?

L370-379 The absence of RNA splicing abnormalities in the recessive RNU2-2 is intriguing. Did the authors consider the alternative that some cases are false positives?

Figure 3D: Family 21 has 2 different combinations of compound heterozygotes in siblings P27 and P28 (one allele the same but the second different). Something is wrong there.

Figure 2B: I suggest to add which nucleotides are different between RNU2-2 from RNU2-1.

* Appropriate use of statistics and treatment of uncertainties

Wondering if the ACMG criteria for pathogenicity are appropriate for non-coding genes.

Is it bothersome to the authors that in Figure 1A the only significant point was the RNU2-2 homozygous?

*Conclusions: robustness, validity, reliability

OK

*Suggested improvements: experiments, data for possible revision

Since this study deals with a recessive phenotype, I suggest that the authors divide the cohort population in "outbred" and consanguineous" and provide frequency statistics for each cohort separately. It is possible that the frequency in inbred NND is substantially higher than in outbred families.

I suggest in the discussion the authors could mention a functional assessment of variants with a method similar to that presented in the preprint of saturation genome editing:

medRxiv 2025.04.08.25325442; doi: <https://doi.org/10.1101/2025.04.08.25325442>

I also suggest to mention in the Discussion the need for model organism experiments including in mice (please mention the evolutionary conservation of RNU2-2 in mice).

In addition, I suggest to include in the discussion that RNU2-2 needs to be included in the Extended Carrier Screening programs.

Important point for the Discussion: Why the heterozygous parents with a variant in n.3, n.5 and other nucleotides in regions n1-n.67 and n.97-n.107 are not symptomatic similar to the dominant RNU2-2?

Could the phenotypic heterogeneity be partially due to the copy number variability of RNU2-1 genes on chromosome 17?

* References: appropriate credit to previous work?

Please refer to other investigations (in the Discussion) that recently reported similar findings.

*Clarity and context: lucidity of abstract/summary, appropriateness of abstract, introduction and conclusions

OK

Reviewer #2 (Remarks to the Author):

A. Summary of the key results: good

B. Originality and significance: if not novel, please include reference: satisfied

C. Data & methodology: validity of approach, quality of data, quality of presentation: good

D. Appropriate use of statistics and treatment of uncertainties: good

E. Conclusions: robustness, validity, reliability: satisfied

F. Suggested improvements: experiments, data for possible revision: good

G. References: appropriate credit to previous work?: satisfied

H. Clarity and context: lucidity of abstract/summary, appropriateness of abstract, introduction and conclusions: good

This report presents an intriguing study on a recessive neurodevelopmental disorder associated with the RNU2-2 gene, caused by biallelic rare variants. The authors reveal that rare biallelic variants in RNU2-2 are enriched and more frequently transmitted in a cohort of individuals with unresolved neurodevelopmental disorders (NDDs). They achieve this by comparing genome sequencing data from these individuals with large population datasets, which helps prioritize candidate disease-causing variants and illustrates the unique genetic architecture of the disorder.

By combining statistical methods with detailed clinical phenotyping, the authors provide compelling evidence that these variants lead to a newly identified recessive developmental and epileptic encephalopathy (DEE). Transcriptomic data from affected individuals indicate that candidate variants are linked to reduced U2-2 transcript levels and a decreased U2-2:U2-1 transcript ratio, suggesting a loss-of-function mechanism.

While the paper is well-written, several concerns have been raised:

1. Is it possible to classify the severity of the disorder by combining biallelic variant combinations with clinical symptoms? Could there be lethal combinations among these variants?
2. Were there any medications that proved particularly effective in treating epileptic seizures associated with this disorder?
3. Is it feasible to link the genes most affected by gene expression changes to the mechanisms of action of the most effective therapeutic agents?

Reviewer #3 (Remarks to the Author):

The authors provide a very well written and significant, original manuscript describing a distinct recessive genetic disorder associated with bi-allelic RNU2-2 variants primarily manifest as epileptic encephalopathy. They hypothesize reduced transcript stability as a pathomechanism and suggest a potential diagnostic biomarker. The data, methodology, conclusions and clarity are all appropriate. I suggest only minor edits primarily for reader clarity:

1. This sentence – "In a rare conditions cohort subject to recruitment bias, variants pathogenic for a recessive condition may be over-transmitted to affected", though correct, may not be easily understandable by all readers, especially a clinical audience. I am assuming the meaning is that recruitment bias such as more frequent inclusion in a cohort of affected sib-pairs or more severe cases may lead to greater than expected representation of transmitted pathogenic variants. If this is correct, it would be helpful to add a brief sentence explaining this concept more clearly.
2. The paragraph referencing the Goldilocks phenomenon, similarly, could include a bit more explanation for readers not familiar with this concept. For example, the sentence "In this scenario, alleles which are pathogenic in the homozygous state may have intermediate deleteriousness." could be adjusted to "In this scenario, alleles which are pathogenic in the homozygous state may have intermediate deleteriousness and thus result in a mild phenotype in the heterozygous state."
3. The authors hypothesize "It is therefore plausible that U2-1 may partially compensate for under- expression of U2-2." Though sample size may limit analysis, it might be informative to evaluate if individuals with bi-allelic RNU2-2 variants WITHOUT NDD might have rare RNU2-1 variants that theoretically may compensate for deleterious RNU2-2 variants. I recognize this analysis is unlikely to be revealing but may be worthwhile to consider.

Version 1:

Decision Letter:

Our ref: NG-A70235R

18th December 2025

Dear Sid,

Your revised manuscript "Biallelic variants in RNU2-2 cause a remarkably frequent developmental and epileptic encephalopathy" (NG-A70235R) has been seen by Reviewer #1, who is satisfied with the revision and has no further requests. (Reviewer #1 provided only Remarks to the Editor at this round of review.) Based on this positive feedback, we will be happy in principle to publish your study in Nature Genetics as an Article pending final revisions to comply with our editorial and formatting guidelines.

We are now performing detailed checks on your paper, and we will send you a checklist detailing our editorial and formatting requirements soon. Please do not upload the final materials or make any revisions until you receive this additional information from us.

Thank you again for your interest in Nature Genetics. Please do not hesitate to contact me if you have any questions.

Sincerely,
Kyle

Kyle Vogan, PhD
Senior Editor
Nature Genetics
<https://orcid.org/0000-0001-9565-9665>

Dear Dr Vogan,

Subject: Manuscript NG-A70235: "Biallelic variants in RNU2-2 cause a remarkably frequent developmental epileptic encephalopathy"

Thank you for considering our study for publication in Nature Genetics. We thank all the reviewers for recognising the importance of our work and their constructive criticism. A point-wise rebuttal to all the points raised by the referees is provided below.

Reviewer #1

1. The conclusions of the study are novel; however, the authors should acknowledge that similar conclusions have been published in preprints from other groups: medRxiv 2025.08.26.25334179; doi: <https://doi.org/10.1101/2025.08.26.25334179> [doi.org] medRxiv 2025.09.02.25334923; doi: <https://doi.org/10.1101/2025.09.02.25334923> [doi.org]. If some conclusions are different from those of the other studies, please mention them in the discussion. We have added the references to the pre-print (**line 473**).
2. Please better clarify the cohorts of patients used, the number of individuals versus number of families with biallelic variants, and the filtering in the different steps, particularly in the discovery cohort. Please also provide more info on the phenotypes in the validation cohorts and the filtering that led to the final 14 additional families. The authors tried to clarify the discovery flow in figure 1C, but I confess its content is not clear. I suggest to rework this figure and its legend. Thanks for the suggestions. **Supplementary Table 3** in our original submission provided the information about variants / families / filtering in the discovery cohort. We have updated the Methods section and also clarified the phenotypes of the cohorts in the Results section (**Line 262-380**). **Figure 1C** has also been updated to improve clarity.
3. An important point: Please mention if the same patient cohorts have been used by the authors of the competing and published (in preprints) studies.
Thank you for the suggestion. Added (**Line 473**).
4. L229-238 not very clear. What was the rationale in L232-233? (For compound heterozygous variants, we only included those genotypes in which at least one variant fell in the 5' constrained region (n.1-n.67) or the Sm site (n.97-n.107)). Is it possible that some true positives were filtered out?
We explained the rationale for our selection criteria for compound heterozygous genotypes in the last paragraph of the previous Results sub-section (The distribution of candidate variants is distinct from controls) in the original submission – "...we show that a strong enrichment of genotypes with at least one variant within n.1 to n.67 in individuals with unsolved NDD versus both 100kGP controls (OR=34.0, 95%CI 6.81-170, two-tailed Fisher's exact P=1.07x10⁻⁷) and 200,011 individuals in UKB for whom statistically phased genome sequencing data was available (OR 153, 95%CI 19.8-1180, P=1.90x10⁻¹⁰). In other recessive RNU-opathies, variants in the Sm binding site are known to be pathogenic." We have now added a sentence to clarify the link in the revised manuscript (**Line 264-267**). The possibility of our search criteria resulting in some true positives being missed cannot be ruled out. We have added a sentence to emphasise this (**Lines 476-479**).
5. L246 31? there are 38 in L226
As mentioned in the next sentence in the original submission – "We excluded seven siblings from this analysis to prevent confounding due to relatedness."
6. L293 delete RNU2-2
Thanks for spotting the typo. Now deleted.

7. **L292 Why 6 individuals without RNU2-2 biallelic variants were outliers for U2-2 RNA levels? Hypotheses?**
U2-2 expression in blood is highly variable with >1000 times difference between lowest and highest level of expression in controls (**Fig 4a**). We suspect that the six outliers in controls for U2-2 RNA levels likely represent the lower extreme of the expression distribution. Importantly, these six individuals also had preserved U2-2:U2-1 ratio (**Fig 4b**). We have added this explanation in the Discussion (**Line 526-527**).
8. **L323-324 The argument that the ratio U2-2/U2-1 is a better marker than the U2-2 RNA abundance is not convincing. Figures 4A and 4C do not seem to support this. Please see response to Point #7.**
9. **Why the RNA data were not normalized to the sum of reads for chromosomes 11 and 17 (or to the whole transcriptome)?**
The RNA data were normalised to the sum of reads for the respective chromosomes. We have updated the legend for clarity.
10. **L370-379 The absence of RNA splicing abnormalities in the recessive RNU2-2 is intriguing. Did the authors consider the alternative that some cases are false positives?**
We highlighted this observation in the Discussion in the original submission – *“Because variants which grossly disrupt splicing are likely to be incompatible with life, any splicing defect in the recessive disorder is likely to be subtle and/or tissue specific. We are unable to comment on the effect on splicing in neural tissue as our analysis is limited to short-read RNA-Seq in whole blood.”* As we demonstrate significantly decreased U2-2:U2-1 ratios in 8/9 of these individuals (c.f. 5,443 controls) it is highly unlikely that these are false positive cases. It is possible that the one case without depleted ratio is false positive, however, this is unlikely to affect the outlier results across the cohort.
11. **Figure 3D: Family 21 has 2 different combinations of compound heterozygotes in siblings P27 and P28 (one allele the same but the second different). Something is wrong there.**
These two children have the same mother but two different fathers. We have added information about all unusual pedigrees with detailed clinical information in new **supplemental figure 6** for clarity.
12. **Figure 2B: I suggest to add which nucleotides are different between RNU2-2 from RNU2-1.**
Thank you for the suggestion. Please see new **supplemental figure 13**.
13. **Wondering if the ACMG criteria for pathogenicity are appropriate for non-coding genes.**
We agree that the ACMG criteria need to be updated for assessing pathogenicity for non-coding genes.
14. **Is it bothersome to the authors that in Figure 1A the only significant point was the RNU2-2 homozygous?**
As shown in Figure 2, the position of the combination of *RNU2-2* compound heterozygous variants is critical in determining their pathogenicity. The information about the variant combinations was not used in the initial enrichment experiments. It is remarkable that even without including position information we detected nominal significance for compound heterozygous variants in this experiment. As mentioned in response to point #3, including the position information gave us highly significant odds ratios.
15. **Since this study deals with a recessive phenotype, I suggest that the authors divide the cohort population in "outbred" and consanguineous" and provide frequency statistics for each cohort separately. It is possible that the frequency in inbred NND is substantially higher than in outbred families.**
Thank you for the excellent suggestion. We agree that determining the carrier and affected frequencies in different populations is important. However, doing justice to this important

question will require a follow-up epidemiologically focussed manuscript. We are working on this question and expect the results to take several months.

16. I suggest in the discussion the authors could mention a functional assessment of variants with a method similar to that presented in the preprint of saturation genome editing: medRxiv 2025.04.08.25325442; doi: <https://doi.org/10.1101/2025.04.08.25325442> [doi.org]

Thank you for the suggestion. Added (**Line 505**).

17. I also suggest to mention in the Discussion the need for model organism experiments including in mice (please mention the evolutionary conservation of RNU2-2 in mice)-

Thank you for the suggestion. Added (**Line 552-553**). **Extended Data Figure 14** shows the truncation of the orthologous U2 copy in mice upstream of WDR74. Further work is required to discover whether mice rely entirely on their RNU2-1 repeat array for U2 snRNA production or whether there is another copy elsewhere in the mouse genome which has taken over the function of RNU2-2 in humans.

18. In addition, I suggest to include in the discussion that RNU2-2 needs to be included in the Extended Carrier Screening programs.

Thank you for the suggestion. We had mentioned preconception screening our discussion, we have changed it to extended carrier screening programs (**Line 558**).

19. Important point for the Discussion: Why the heterozygous parents with a variant in n.3, n.5 and other nucleotides in regions n1-n.67 and n.97-n.107 are not symptomatic similar to the dominant RNU2-2?

The dominant *RNU-2* variants are n.3C>A, n.4G>A, n.5C>A, n.7_9insA, n.35A>G and n.35A>C (See figure below taken from Jackson *et al* Nat Genet 2025).

Of note, none of the heterozygous carrier parents of children with the recessive RNU2-2 syndrome have these exact dominant variants. Specifically, we found carrier parents with (c.3C>T and n.5C>T) different substitutions at the same positions. With regards to why heterozygous parents are not affected – we think that the disease mechanism of the dominant and recessive diseases are different (**see Figure 4**). Further studies will be required to understand the underlying mechanisms, especially of the dominant syndrome.

20. Could the phenotypic heterogeneity be partially due to the copy number variability of RNU2-1 genes on chromosome 17?

Thank you for the suggestion. We have not explored his hypothesis as reliably determining the copy number of RNU2-1 from short read genome sequencing data is challenging.

Reviewer #2

21. **Is it possible to classify the severity of the disorder by combining biallelic variant combinations with clinical symptoms? Could there be lethal combinations among these variants**
Thank you for the excellent suggestion. A detailed genotype-phenotype correlation will be part of a follow-up study. We agree that some combinations may be lethal. We raised this possibility in the second paragraph of our Discussion of our original submission (**line 490-491**).
22. **Were there any medications that proved particularly effective in treating epileptic seizures associated with this disorder?**
Thank you for raising this important point. We have not identified a particularly effective drug unfortunately (**Supp Table 7**).
23. **Is it feasible to link the genes most affected by gene expression changes to the mechanisms of action of the most effective therapeutic agents?**
Unfortunately, we did not identify significant gene expression changes in the recessive disease (**Fig 5A**).

Reviewer #3

24. **This sentence – "In a rare conditions cohort subject to recruitment bias, variants pathogenic for a recessive condition may be over-transmitted to affected", though correct, may not be easily understandable by all readers, especially a clinical audience. I am assuming the meaning is that recruitment bias such as more frequent inclusion in a cohort of affected sib-pairs or more severe cases may lead to greater than expected representation of transmitted pathogenic variants. If this is correct, it would be helpful to add a brief sentence explaining this concept more clearly.**
Thanks for the suggestion. We have reworded the sentence (Line 179-181).
25. **The paragraph referencing the Goldilocks phenomenon, similarly, could include a bit more explanation for readers not familiar with this concept. For example, the sentence "In this scenario, alleles which are pathogenic in the homozygous state may have intermediate deleteriousness." could be adjusted to "In this scenario, alleles which are pathogenic in the homozygous state may have intermediate deleteriousness and thus result in a mild phenotype in the heterozygous state."**
Thanks for the suggestion. We have reworded the section (Line 484-491).
26. **The authors hypothesize "It is therefore plausible that U2-1 may partially compensate for under-expression of U2-2." Though sample size may limit analysis, it might be informative to evaluate if individuals with bi-allelic RNU2-2 variants WITHOUT NDD might have rare RNU2-1 variants that theoretically may compensate for deleterious RNU2-2 variants. I recognize this analysis is unlikely to be revealing but may be worthwhile to consider.**
Thanks for the interesting idea. RNU2-1 occurs in a variable repeat array of identical copies. Hence, accurate identification of *RNU2-1* variants is extremely challenging with short read sequencing data.